# Unleash the Potential of Image Branch for Cross-modal 3D Object Detection

**Yifan Zhang**[1], **Qijian Zhang**[1], **Junhui Hou**[1]*, **Yixuan Yuan**[2]*, and **Guoliang Xing**[2]

[1]City University of Hong Kong, [2]The Chinese University of Hong Kong

{yzhang3362-c,qijizhang3-c}@my.cityu.edu.hk;jh.hou@cityu.edu.hk;
yxyuan@ee.cuhk.edu.hk;glxing@ie.cuhk.edu.hk

## Abstract

To achieve reliable and precise scene understanding, autonomous vehicles typically incorporate multiple sensing modalities to capitalize on their complementary attributes. However, existing cross-modal 3D detectors do not fully utilize the image domain information to address the bottleneck issues of the LiDAR-based detectors. This paper presents a new cross-modal 3D object detector, namely UPIDet, which aims to **u**nleash the **p**otential of the **i**mage branch from two aspects. First, UPIDet introduces a new 2D auxiliary task called normalized local coordinate map estimation. This approach enables the learning of local spatial-aware features from the image modality to supplement sparse point clouds. Second, we discover that the representational capability of the point cloud backbone can be enhanced through the gradients backpropagated from the training objectives of the image branch, utilizing a succinct and effective point-to-pixel module. Extensive experiments and ablation studies validate the effectiveness of our method. Notably, we achieved the top rank in the highly competitive cyclist class of the KITTI benchmark at the time of submission. The source code is available at https://github.com/Eaphan/UPIDet.

## 1 Introduction

In recent years, there has been increasing attention from both academia and industry towards 3D object detection, particularly in the context of autonomous driving scenarios [11, 57, 56, 58]. Two dominant data modalities - 3D point clouds and 2D RGB images - demonstrate complementary properties, with point clouds encoding accurate structure and depth cues but suffering from sparsity, incompleteness, and non-uniformity. On the other hand, RGB images convey rich semantic features and has already developed powerful learning architectures [10], but pose challenges in reliably modeling spatial structures. Despite the efforts made by previous studies [43, 40] devoted to cross-modal learning, how to reasonably mitigate the substantial gap between two modalities and utilize the information from the image domain to complement single-modal detectors remains an open and significant issue.

Currently, the performance of single-modal detectors is limited by the sparsity of observed point clouds. As shown in Figure 1 (b), the performance of detectors on objects with fewer points (LEVEL_2) is significantly lower than that on objects with denser points (LEVEL_1). And Fig.1 (a) shows the distribution of point clouds in a single object for the Waymo dataset. It can be observed that a substantial proportion of objects have less than 50 point clouds, making it challenging to deduce the 3D bounding boxes with seven degrees of freedom from very sparse point clouds, as discussed in detail in Sec.3. While the observed point cloud can be sparse due to heavy occlusion or low reflectivity material, their contour and appearance can still be clear in RGB images (as shown in the example in Fig.1 (c)). Despite previous work exploring the use of semantic image features to enhance features extracted from LiDAR [40, 12], they have not fundamentally improved the spatial

---

*Corresponding author.

37th Conference on Neural Information Processing Systems (NeurIPS 2023).

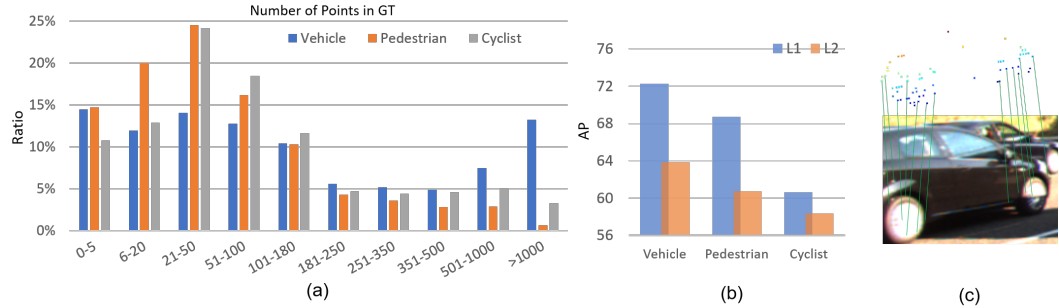

Figure 1: (a) Distribution of number of points in a single object in Waymo Dataset.(b) Performance of the Second on Waymo [38]. (c) Point cloud and image of a black car and part of their correspondence.

representation limited by the sparsity of the point cloud. There is a class of work that attempts to generate pseudo LiDAR points as a supplement to sparse point clouds but these models commonly rely on depth estimators trained on additional data [43].

On the other hand, the performance of LiDAR-based detection frameworks is limited by the inherent challenges of processing *irregular* and *unordered* point clouds, which makes it difficult for neural network backbones to learn effectively from 3D LiDAR data. Consequently, the representation capability of these backbone networks is still relatively insufficient, despite the performance boost achieved by previous cross-modal methods[12, 40] that incorporate discriminative 2D semantic information into the 3D detection pipeline. However, there is no evidence that these methods actually enhance the representation capability of the 3D LiDAR backbone network, which is also one of the most critical influencing factors.

This paper addresses the first issue mentioned above by leveraging local-spatial aware information from RGB images. As depicted in Fig. 1 (c), predicting the 3D bounding box of a reflective black car from the sparse observed point cloud is challenging. By incorporating image information, we can not only distinguish the foreground points but also determine the relative position of the point cloud within the object, given the known 3D-to-2D correspondence. To learn local spatial-aware feature representations, we introduce a new 2D auxiliary task called normalized local coordinate (NLC) map estimation, which provides the relative position of each pixel inside the object. Ultimately, we expect to improve the performance of the cross-modal detector by utilizing both the known *global* coordinates of points and their estimated *relative* position within an object.

On the other hand, the depth of point cloud serves as a complementary source to RGB signals, providing 3D geometric information that is resilient to lighting changes and aids in distinguishing between various objects. In light of this, we present a succinct point-to-pixel feature propagation module that allows 3D geometric features extracted from LiDAR point clouds to flow into the 2D image learning branch. Unlike the feature-level enhancements brought by existing pixel-to-point propagation, this method enhances the representation ability of 3D LiDAR backbone networks. Remarkably, this is an interesting observation because the back-propagated gradients from the 2D image branch built upon established 2D convolutional neural networks (CNNs) with impressive learning ability can effectively boost the 3D LiDAR branch.

We conduct extensive experiments on the prevailing KITTI [8] benchmark dataset. Compared with state-of-the-art single-modal and cross-modal 3D object detectors, our framework achieves remarkable performance improvement. Notably, our method ranks $1^{st}$ on the cyclist class of KITTI 3D detection benchmark[2]. Conclusively, our main contributions are three-fold:

- We demonstrate that the representational capability of the point cloud backbone can be enhanced through the gradients backpropagated from the training objectives of the image branch, utilizing a succinct and effective point-to-pixel module.

- We reveal the importance of specific 2D auxiliary tasks used for training the 2D image learning branch which has not received enough attention in previous works, and introduce 2D NLC

---

[2]www.cvlibs.net/datasets/kitti/eval_object.php?obj_benchmark=3d

map estimation to facilitate learning spatial-aware features and implicitly boost the overall 3D detection performance.

- For the first time, we provide a rigorous analysis of the performance bottlenecks of single-modal detectors from the perspective of degrees of freedom (DOF) of the 3D bounding box.

## 2 Related Work

**LiDAR-based 3D Object Detection** could be roughly divided into two categories. *1) Voxel-based 3D detectors* typically voxelize the point clouds into grid-structure forms of a fixed size [61]. [47] introduced a more efficient sparse convolution to accelerate training and inference. *2) Point-based 3D detectors* consume the raw 3D point clouds directly and generate predictions based on (downsampled) points. [32] applied a point-based feature extractor and generated high-quality proposals on foreground points. 3DSSD [49] adopted a new sampling strategy named F-FPS as a supplement to D-FPS to preserve enough interior points of foreground instances. It also built a one-stage anchor-free 3D object detector based on feasible representative points. [55] and [2] introduced semantic-aware down-sampling strategies to preserve foreground points as much as possible. [36] proposed to encode the point cloud by a fixed-radius near-neighbor graph.

**Camera-based** 3D Object Detection. Early works [25, 24] designed monocular 3D detectors by referring to 2D detectors [31, 39] and utilizing 2D-3D geometric constraints. Another way is to convert images to pseudo-lidar representations via monocular depth estimation and then resort to LiDAR-based methods [43]. [3] built dense 2D-3D correspondence by predicting normalized object coordinate map [42] and adopted uncertainty-driven PnP to estimate object location. Our proposed NLC map estimation differs in that it aims to extract local spatial features from image features and does not rely on labeled dense depth maps and estimated RoI boxes. Recently, multi-view detectors like BEVStereo [17] have also achieved promising performance, getting closer to LiDAR-based methods.

**Cross-Modal 3D Object Detection** can be roughly divided into four categories: proposal-level, result-level, point-level, and pseudo-lidar-based approaches. *Proposal-level* fusion methods [5, 14, 19] adopted a two-stage framework to fuse image features and point cloud features corresponding to the same anchor or proposal. For *result-level* fusion, [26] exploited the geometric and semantic consistencies of 2D detection results and 3D detection results to improve single-modal detectors. Obviously, both proposal-level and decision-level fusion strategies are coarse and do not make full use of correspondence between LiDAR points and images. *Pseudo-LiDAR* based methods [52, 45, 19] employed depth estimation/completion and converted the estimated depth map to pseudo-LiDAR points to complement raw sparse point clouds. However, such methods require extra depth map annotations that are high-cost. Regarding *point-level* fusion methods, [18] proposed the continuous fusion layer to retrieve corresponding image features of nearest 3D points for each grid in BEV feature map. Especially, [40] and [12] retrieved the semantic scores or features by projecting points to image plane. However, neither semantic segmentation nor 2D detection tasks can enforce the network to learn 3D-spatial-aware image features. By contrast, we introduce NLC Map estimation as an auxiliary task to promote the image branch to learn local spatial information to supplement the sparse point cloud. Besides, [1] proposed TransFusion, a transformer-based detector that performs fine-grained fusion with attention mechanisms on the BEV level. [27] and [20] estimated the depth of multi-view images and transformed the camera feature maps into the BEV space before fusing them with LiDAR BEV features.

## 3 Preliminary

This section introduces the definition of the NLC map and analyzes the performance bottlenecks of single-modal detectors in terms of degrees of freedom (DOF). The goal of the cross-modal 3D detector is to estimate the bounding box parameters, including the object dimensions $(w, l, h)$, center location $(x_c, y_c, z_c)$, and orientation angle $\theta$. The input consists of an RGB image $\boldsymbol{X} \in \mathbb{R}^{3 \times H \times W}$, where $H$ and $W$ denote the height and width of image, respectively. Additionally, the input includes a 3D LiDAR point cloud $\boldsymbol{p}_i | i = 1, ..., N$, where $N$ is the number of points, and each point $\boldsymbol{p}_i \in \mathbb{R}^4$ is defined by its 3D location $(x_p, y_p, z_p)$ in the LiDAR coordinate system and the reflectance value $\rho$.

**Normalized Local Coordinate (NLC) System.**
We define the NLC system as follows: we take the center of an object as the origin, align the $x$-axis towards the head direction of its ground-truth (GT) bounding box, and then normalize the local coordinates with respect to the size of the GT bounding box [7, 34]. Figure 2 (a) shows an example of the NLC system for a typical car. With the geometry relationship between the input RGB image and point cloud, we can project the NLCs to the 2D

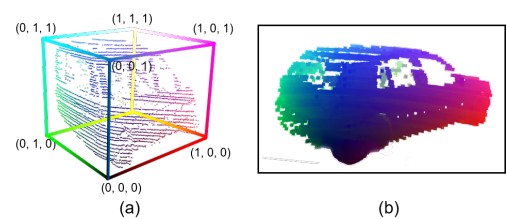

Figure 2: Illustration of the (a) NLC system and (b) 2D NLC map.

image plane and construct a 2D NLC map with three channels corresponding to the three spatial dimensions, as illustrated in Fig.2(b).

Specifically, to build the correspondence between the 2D and 3D modalities, we project the observed points from the 3D coordinate system to the 2D coordinate system on the image plane:

$$d_c\,[u\ v\ 1]^T = \boldsymbol{K}\,[\boldsymbol{R}\quad \boldsymbol{T}]\,[x_p\ y_p\ z_p\ 1]^T , \tag{1}$$

where $u$, $v$, $d_c$ denote the corresponding coordinates and depth on the image plane, $\boldsymbol{R} \in \mathbb{R}^{3\times3}$ and $\boldsymbol{T} \in \mathbb{R}^{3\times1}$ denote the rotation matrix and translation matrix of the LiDAR relative to the camera, and $\boldsymbol{K} \in \mathbb{R}^{3\times3}$ is the camera intrinsic matrix.

Given the point cloud and the bounding box of an object, LiDAR coordinates of foreground points can be transformed into the NLC system by proper translation, rotation, and scaling, i.e.,

$$\begin{bmatrix}x_p^{NLC}\\y_p^{NLC}\\z_p^{NLC}\end{bmatrix} = \begin{bmatrix}1/l & 0 & 0\\0 & 1/w & 0\\0 & 0 & 1/h\end{bmatrix}\cdot\begin{bmatrix}x_p^{LCS}\\y_p^{LCS}\\z_p^{LCS}\end{bmatrix}+\begin{bmatrix}0.5\\0.5\\0.5\end{bmatrix} = \begin{bmatrix}1/l & 0 & 0\\0 & 1/w & 0\\0 & 0 & 1/h\end{bmatrix}\cdot\begin{bmatrix}cos\theta & -sin\theta & 0\\sin\theta & cos\theta & 0\\0 & 0 & 1\end{bmatrix}\cdot\begin{bmatrix}x_p-x_c\\y_p-y_c\\z_p-z_c\end{bmatrix}+\begin{bmatrix}0.5\\0.5\\0.5\end{bmatrix}, \tag{2}$$

where $(x_p^{NLC}, y_p^{NLC}, z_p^{NLC})$ and $(x_p^{LCS}, y_p^{LCS}, z_p^{LCS})$ denote the coordinates of points in NLC system and local coordinate system, respectively.

**Discussion.** Suppose that the bounding box is unknown, but the global coordinates and NLCs of points are known. We can build a set of equations to solve for the parameters of the box, namely $x_c, y_c, z_c, w, l, h, \theta$. To solve for all seven parameters, we need at least seven points with known global coordinates and NLCs to construct the equations. However, as shown in Fig. 1, the number of points for some objects is too small to estimate the 3D box with 7 degrees of freedom. Moreover, it is challenging to infer NLCs of points with only LiDAR data for objects far away or with low reflectivity, as the point clouds are sparse. However, RGB images can still provide visible contours and appearances for these cases. Therefore, we propose to estimate the NLCs of observed points based on enhanced image features. We can then retrieve the estimated NLCs based on the 2D-3D correspondence. Finally, we expect the proposed cross-modal detector to achieve higher detection accuracy with estimated NLCs and known global coordinates.

## 4 Proposed Method

**Overview**. Architecturally, the processing pipeline for cross-modal 3D object detection includes an image branch and a point cloud branch, which learn feature representations from 2D RGB images and 3D LiDAR point clouds, respectively. Most existing methods only incorporate pixel-wise semantic cues from the 2D image branch into the 3D point cloud branch for feature fusion. However, we observe that point-based networks typically show insufficient learning ability due to the essential difficulties in processing irregular and unordered point cloud data modality [53]. Therefore, we propose to boost the expressive power of the point cloud backbone network with the assistance of the 2D image branch. Our ultimate goal is not to design a more powerful point cloud backbone network structure, but to make full use of the available resources in this cross-modal application scenario.

As shown in Fig.3, in addition to pixel-to-point feature propagation explored by mainstream point-level fusion methods[18, 12], we employ point-to-pixel propagation allowing features to flow inversely from the point cloud branch to the image branch. We achieve bidirectional feature propagation in a multi-stage fashion. In this way, not only can image features propagated via the pixel-to-point module provide additional information, but gradients backpropagated from the training objectives of the image branch can boost the representation ability of the point cloud backbone. We also employ auxiliary tasks to train the pipeline, aiming to enforce the network to learn rich semantic and spatial

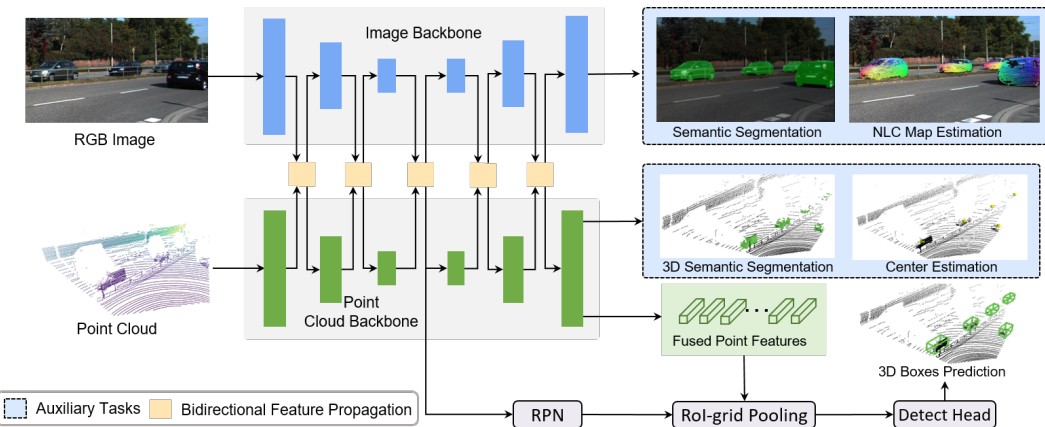

Figure 3: Flowchart of the proposed UPIDet for cross-modal 3D object detection. It contains two branches separately extracting features from input 2D RGB images and 3D point clouds. UPIDet is mainly featured with (1) the point-to-pixel module that enables gradient backpropagation from 2D branch to 3D branch and (2) the supervision task of NLC map estimation for promoting the image branch to exploit local spatial features.

representations. In particular, we propose NLC map estimation to promote the image branch to learn spatial-aware features, providing a necessary complement to the sparse spatial representation extracted from point clouds, especially for distant or highly occluded cases. Appendix A.1 provides detailed network structures of the image and point cloud backbones.

## 4.1 Bidirectional Feature Propagation

The proposed bidirectional feature propagation consists of a pixel-to-point module and a point-to-pixel module, which bridges the two learning branches that operate on RGB images and LiDAR point clouds. Functionally, the point-to-pixel module applies grid-level interpolation on point-wise features to produce a 2D feature map, while the pixel-to-point module retrieves the corresponding 2D image features by projecting 3D points to the 2D image plane.

In general, we partition the overall 2D and 3D branches into the same number of stages. This allows us to perform bidirectional feature propagation at each stage between the corresponding layers of the image and point cloud learning networks. Without loss of generality, we focus on a certain stage where we can acquire a 2D feature map $\boldsymbol{F} \in \mathbb{R}^{C_{2D} \times H' \times W'}$ with $C_{2D}$ channels and dimensions $H' \times W'$ from the image branch and a set of $C_{3D}$-dimensional embedding vectors $\boldsymbol{G} = \left\{ \boldsymbol{g}_i \in \mathbb{R}^{C_{3D}} \right\}_{i=1}^{N'}$ of the corresponding $N'$ 3D points.

**Point-to-Pixel Module.** As illustrated in Fig. 4, we design the point-to-pixel module to propagate geometric information from the 3D point cloud branch into the 2D image branch. Formally, we

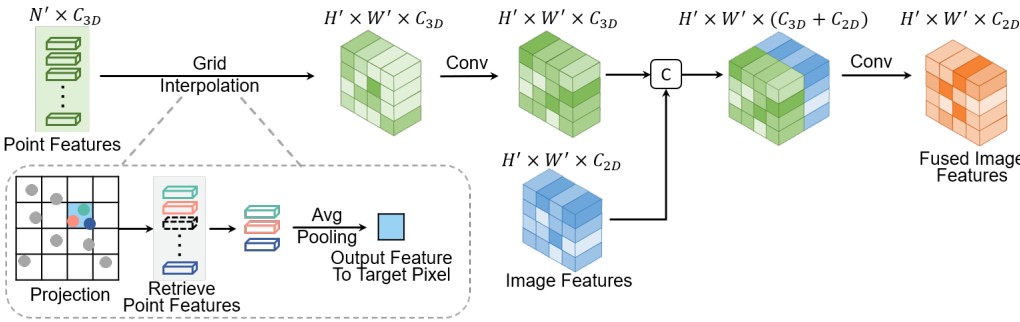

Figure 4: Illustration of the proposed point-to-pixel module.

aim to construct a 2D feature map $\boldsymbol{F}^{P2I} \in \mathbb{R}^{C_{3D} \times H' \times W'}$ by querying from the acquired point-wise embeddings $\boldsymbol{G}$. More specifically, for each pixel position $(r, c)$ over the targeted 2D grid of $\boldsymbol{F}^{P2I}$ with dimensions $H' \times W'$, we collect the corresponding geometric features of points that are projected inside the range of this pixel and then perform feature aggregation by avg-pooling $\mathtt{AVG}(\cdot)$:

$$\boldsymbol{F}_{r,c}^{P2I} = \mathtt{AVG}(\boldsymbol{g}_j | j = 1, ..., n), \tag{3}$$

where $n$ counts the number of points that can be projected inside the range of the pixel located at the $r^{th}$ row and the $c^{th}$ column. For empty pixel positions where no points are projected inside, we define its feature value as $\boldsymbol{0}$.

After that, instead of directly feeding the "interpolated" 2D feature map $\boldsymbol{F}^{P2I}$ into the subsequent 2D convolutional stage, we introduce a few additional refinement procedures for feature smoothness as described below:

$$\boldsymbol{F}^{fuse} = \mathtt{CONV}(\mathtt{CAT}[\mathtt{CONV}(\boldsymbol{F}^{P2I}), \boldsymbol{F}]), \tag{4}$$

where $\mathtt{CAT}[\cdot, \cdot]$ and $\mathtt{CONV}(\cdot)$ stand for feature channel concatenation and the convolution operation, respectively. The resulting $\boldsymbol{F}^{fuse}$ is further fed into the subsequent layers of the 2D image branch.

**Pixel-to-Point Module.** This module propagates semantic information from the 2D image branch into the 3D point cloud branch. We project the corresponding $N'$ 3D points onto the 2D image plane, obtaining their planar coordinates denoted as $\boldsymbol{X} = \{\boldsymbol{x}_i \in \mathbb{R}^2\}_{i=1}^{N'}$. Considering that these 2D coordinates are continuously and irregularly distributed over the projection plane, we apply bilinear interpolation to compute exact image features at projected positions, deducing a set of point-wise embeddings $\boldsymbol{F}^{I2P} = \{\boldsymbol{F}(\boldsymbol{x}_i) \in \mathbb{R}^{C_{2D}} | i = 1, ..., N'\}$. Similar to our practice in the point-to-pixel module, the interpolated 3D point-wise features $\boldsymbol{F}^{I2P}$ need to be refined by the following procedures:

$$\boldsymbol{G}^{fuse} = \mathtt{MLP}\left(\mathtt{CAT}\left[\mathtt{MLP}\left(\boldsymbol{F}^{I2P}; \boldsymbol{\theta}_1\right), \boldsymbol{G}\right]; \boldsymbol{\theta}_2\right), \tag{5}$$

where $\mathtt{MLP}(\cdot; \boldsymbol{\theta})$ denotes shared multi-layer perceptrons parameterized by learnable weights $\boldsymbol{\theta}$, and the resulting $\boldsymbol{G}^{fuse}$ further passes through the subsequent layers of the 3D point cloud branch.

*Remark.* Unlike some prior work that introduces complicated bidirectional fusion mechanisms [16, 21, 23] to seek more interaction between the two modalities and finally obtain more comprehensive features, we explore a concise point-to-pixel propagation strategy from the perspective of directly improving the representation capability of the 3D LiDAR backbone network. We believe that a concise but effective point-to-pixel propagation strategy will better demonstrate that with feasible 2D auxiliary tasks, the gradients backpropagated from the training objectives of the image branch can boost the representation ability of the point cloud backbone.

## 4.2 Auxiliary Tasks for Training

The semantic and spatial-aware information is essential for determining the category and boundary of objects. To promote the network to uncover such information from input data, we leverage auxiliary tasks when training the whole pipeline. For the point cloud branch, following SA-SSD [9], we introduce 3D semantic segmentation and center estimation to learn structure-aware features. For the image branch, we propose NLC map estimation in addition to 2D semantic segmentation.

**NLC Map Estimation.** Vision-based semantic information is useful for networks to distinguish foreground from background. However, existing detectors face a performance bottleneck due to limited localization accuracy in distant or highly occluded cases where the spatial structure is incomplete due to the sparsity of point clouds. To address this, we use NLC maps as supervision to learn the relative position of each pixel inside the object from the image. This auxiliary task can drive the image branch to learn local spatial-aware features that complement the sparse spatial representation extracted from point clouds. Besides, it can augment the representation ability of the point cloud branch.

*Remark.* The proposed NLC map estimation shares the same objective with pseudo-LiDAR-based methods, which aim to enhance the spatial representation limited by the sparsity of point clouds. However, compared to learning a pseudo-LiDAR representation via depth completion, our NLC map estimation has several advantages: **1)** the *local* NLC is easier to learn than the *global* depth owing to its scale-invariant property at different distances; **2)** our NLC map estimation can be naturally incorporated and trained with the detection pipeline end-to-end, while it is non-trivial to

optimize the depth estimator and LiDAR-based detector simultaneously in pseudo-LiDAR based methods although it can be implemented in a somewhat complex way [30]; and **3)** pseudo-LiDAR representations require ground-truth dense depth images as supervision, which may not be available in reality, whereas our method does not require such information.

**Semantic Segmentation.** Leveraging semantics of 3D point clouds has demonstrated effectiveness in point-based detectors, owing to the explicit preservation of foreground points at downsampling [55, 2]. Therefore, we introduce the auxiliary semantic segmentation tasks not only in the point cloud branch but also in the image branch to exploit richer semantic information. Besides, additional semantic features extracted from images can facilitate the network to distinguish true positive candidate boxes from false positives.

### 4.3 Loss Function

The NLC map estimation task is optimized with the loss function defined as

$$L_{NLC} = \frac{1}{N_{pos}} \sum_i^N \left\| \boldsymbol{p}_i^{NLC} - \hat{\boldsymbol{p}}_i^{NLC} \right\|_H \cdot \mathbf{1}_{\boldsymbol{p}_i}, \tag{6}$$

where $\boldsymbol{p}_i$ is the $i^{th}$ LiDAR point, $N_{pos}$ is the number of foreground LiDAR points, $\|\cdot\|_H$ is the Huber-loss, $\mathbf{1}_{\boldsymbol{p}_i}$ indicates the loss is calculated only with foreground points, and $\boldsymbol{p}_i^{NLC}$ and $\hat{\boldsymbol{p}}_i^{NLC}$ are the NLCs of ground-truth and prediction at corresponding pixel for foreground points.

We use the standard cross-entropy loss to optimize both 2D and 3D semantic segmentation, denoted as $L_{sem}^{2D}$ and $L_{sem}^{3D}$ respectively. And the loss of center estimation is computed as

$$L_{ctr} = \frac{1}{N_{pos}} \sum_i^N \|\Delta \boldsymbol{p}_i - \Delta \hat{\boldsymbol{p}}_i\|_H \cdot \mathbf{1}_{\boldsymbol{p}_i}, \tag{7}$$

where $\Delta \boldsymbol{p}_i$ is the target offsets from points to the corresponding object center, and $\Delta \hat{\boldsymbol{p}}_i$ denotes the output of center estimation head.

Besides the auxiliary tasks, we define the loss of the RPN stage $L_{rpn}$ following [2]. In addition, we also adopt commonly used proposal refinement loss $L_{rcnn}$ as defined in [33]. In all, the loss for optimizing the overall pipeline in an end-to-end manner is written as

$$L_{total} = L_{rpn} + L_{rcnn} + \lambda_1 L_{NLC} + \lambda_2 L_{sem}^{2D} + \lambda_3 L_{sem}^{3D} + \lambda_4 L_{ctr}, \tag{8}$$

where $\lambda_1$, $\lambda_2$, $\lambda_3$, and $\lambda_4$ are hyper-parameters that are empirically set to 1.

## 5 Experiments

### 5.1 Experiment Settings

**Datasets and Metrics.** We conducted experiments on the prevailing KITTI benchmark dataset, which contains two modalities of 3D point clouds and 2D RGB images. Following previous works [33], we divided all training data into two subsets, i.e., 3712 samples for training and the rest 3769 for validation. Performance is evaluated by the Average Precision (AP) metric under IoU thresholds of 0.7, 0.5, and 0.5 for car, pedestrian, and cyclist categories, respectively. We computed APs with 40 sampling recall positions by default, instead of 11. For the 2D auxiliary task of semantic segmentation, we used the instance segmentation annotations as provided in [29]. Besides, we also conducted experiments on the Waymo Open Dataset (WOD) [38], which can be found in Appendix A.3.

**Implementation Details.** For the image branch, we used ResNet18 [10] as the backbone encoder, followed by a decoder composed of pyramid pooling module [59] and several upsampling layers to give the outputs of semantic segmentation and NLC map estimation. For the point cloud branch, we deployed a point-based network like [49] with extra FP layers and further applied semantic-guided farthest point sampling (S-FPS) [2] in SA layers. Thus, we implemented bidirectional feature propagation between the top of each SA or FP layer and their corresponding locations at the image branch. Please refer to Appendix A.1 for more details.

Table 1: Performance comparisons on the KITTI test set, where the best and the second best results are highlighted in bold and underlined, respectively.

| Method | Modality | 3D Car (IoU=0.7) | | | 3D Ped. (IoU=0.5) | | | 3D Cyc. (IoU=0.5) | | | mAP |
|---|---|---|---|---|---|---|---|---|---|---|---|
| | | Easy | Mod. | Hard | Easy | Mod. | Hard | Easy | Mod. | Hard | |
| PointRCNN [32] | LiDAR | 86.96 | 75.64 | 70.70 | 47.98 | 39.37 | 36.01 | 74.96 | 58.82 | 52.53 | 60.33 |
| PointPillars [15] | LiDAR | 82.58 | 74.31 | 68.99 | 51.45 | 41.92 | 38.89 | 77.10 | 58.65 | 51.92 | 60.65 |
| TANet [22] | LiDAR | 84.39 | 75.94 | 68.82 | 53.72 | 44.34 | 40.49 | 75.70 | 59.44 | 52.53 | 61.71 |
| IA-SSD [55] | LiDAR | 88.34 | 80.13 | 75.04 | 46.51 | 39.03 | 35.60 | 78.35 | 61.94 | 55.70 | 62.29 |
| STD [48] | LiDAR | 87.95 | 79.71 | 75.09 | 53.29 | 42.47 | 38.35 | 78.69 | 61.59 | 55.30 | 63.60 |
| Point-GNN [35] | LiDAR | 88.33 | 79.47 | 72.29 | 51.92 | 43.77 | 40.14 | 78.60 | 63.48 | 57.08 | 63.90 |
| Part-$A^2$ [34] | LiDAR | 87.81 | 78.49 | 73.51 | 53.10 | 43.35 | 40.06 | 79.17 | 63.52 | 56.93 | 63.99 |
| PV-RCNN [33] | LiDAR | 90.25 | 81.43 | 76.82 | 52.17 | 43.29 | 40.29 | 78.60 | 63.71 | 57.65 | 64.91 |
| 3DSSD [49] | LiDAR | 88.36 | 79.57 | 74.55 | 54.64 | 44.27 | 40.23 | 82.48 | 64.10 | 56.90 | 65.01 |
| HotSpotNet [4] | LiDAR | 87.60 | 78.31 | 73.34 | 53.10 | 45.37 | 41.47 | 82.59 | 65.95 | 59.00 | 65.19 |
| PDV [11] | LiDAR | **90.43** | 81.86 | 77.36 | 47.80 | 40.56 | 38.46 | 83.04 | 67.81 | 60.46 | 65.31 |
| MV3D [5] | LiDAR+RGB | 74.97 | 63.63 | 54.00 | - | - | - | - | - | - | - |
| MMF [19] | LiDAR+RGB | 88.40 | 77.43 | 70.22 | - | - | - | - | - | - | - |
| AVOD-FPN [14] | LiDAR+RGB | 83.07 | 71.76 | 65.73 | 50.46 | 42.27 | 39.04 | 63.76 | 50.55 | 44.93 | 56.84 |
| F-PointNet [28] | LiDAR+RGB | 82.19 | 69.79 | 60.59 | 50.53 | 42.15 | 38.08 | 72.27 | 56.12 | 49.01 | 57.86 |
| PointPainting [40] | LiDAR+RGB | 82.11 | 71.70 | 67.08 | 50.32 | 40.97 | 37.84 | 77.63 | 63.78 | 55.89 | 60.81 |
| F-ConvNet [44] | LiDAR+RGB | 87.36 | 76.39 | 66.69 | 52.16 | 43.38 | 38.80 | 81.98 | 65.07 | 56.54 | 63.15 |
| CAT-Det [54] | LiDAR+RGB | 89.87 | 81.32 | 76.68 | 54.26 | 45.44 | 41.94 | 83.68 | 68.81 | 61.45 | 67.05 |
| EQ-PVRCNN [50] | LiDAR+RGB | 90.13 | 82.01 | 77.53 | **55.84** | 47.02 | 42.94 | 85.41 | 69.10 | 62.30 | 68.03 |
| UPIDet (Ours) | LiDAR+RGB | 89.13 | **82.97** | **80.05** | 55.59 | **48.77** | **46.12** | **86.74** | **74.32** | **67.45** | **70.13** |

Table 2: Ablative experiments on different feature exploitation and propagation schemes.

| Exp. | Method | 3D Car (IoU=0.7) | | | 3D Ped. (IoU=0.5) | | | 3D Cyc. (IoU=0.5) | | | mAP |
|---|---|---|---|---|---|---|---|---|---|---|---|
| | | Easy | Mod. | Hard | Easy | Mod. | Hard | Easy | Mod. | Hard | |
| (a) | Single-Modal | 91.92 | 85.22 | 82.98 | 68.82 | 61.47 | 56.39 | 91.93 | 74.56 | 69.58 | 75.88 |
| (b) | Point-to-pixel | 91.83 | 85.11 | 82.91 | 70.66 | 63.57 | 58.19 | 92.78 | 76.31 | 71.77 | 77.01 |
| (c) | Pixel-to-point | 92.07 | 85.79 | 82.95 | 69.46 | 65.53 | 59.56 | 94.39 | 75.24 | 72.23 | 77.47 |
| (d) | Bidirectional | 92.63 | 85.77 | 83.13 | 72.68 | 67.64 | 62.25 | 94.39 | 77.77 | 71.47 | 78.64 |

## 5.2 Comparison with State-of-the-Art Methods

We submitted our results to the official KITTI website, and the results are compared in Table 1. Our UPIDet outperforms existing state-of-the-art methods on the KITTI test set by a remarkable margin, with an absolute increase of 2.1% mAP over the second best method EQ-PVRCNN. Notably, at the time of submission, we achieved the top rank on the KITTI 3D detection benchmark for the cyclist class. We observe that UPIDet exhibits consistent and more significant improvements on the "moderate" and "hard" levels, where objects are distant or highly occluded with sparse points. We attribute these performance gains to our bidirectional feature propagation mechanism, which more adequately exploits complementary information between multi-modalities, and the effectiveness of our proposed 2D auxiliary tasks (as verified in Sec.5.3). Furthermore, actual visual results (presented in Appendix A.6) demonstrate that UPIDet produces higher-quality 3D bounding boxes in various scenes.

## 5.3 Ablation Study

We conducted comprehensive ablation studies to validate the effectiveness and explore the impacts of key modules involved in the overall learning framework.

**Effectiveness of Feature Propagation Strategies.** We performed detailed ablation studies on specific multi-modal feature exploitation and interaction strategies. We started by presenting a single-modal baseline (Table 2(a)) that only preserves the point cloud branch of our UPIDet for both training and inference. Based on our complete bidirectional propagation pipeline (Table 2(d)), we explored another two variants as shown in (Table 2(b)) and (Table 2(c)), solely preserving the point-to-pixel and pixel-to-point feature propagation in our 3D detectors, respectively. Note that in Table 2(b), the point-to-pixel feature flow was only enabled during training, and we detached the point-to-pixel module as well as the whole image branch during inference. Empirically, we can draw several conclusions that strongly support our preceding claims. *First*, the performance of Table 2(a) is the worst among all variants, which reveals the superiority of cross-modal learning. *Second*, combining Table 2(a)

Table 3: Ablative experiments on key modules of our UPIDet. LiDAR: the point cloud branch trained without auxiliary tasks; Img: the 2D RGB image backbone; 3D Seg: 3D semantic segmentation; 3D Ctr: 3D center estimation; 2D Seg: 2D semantic segmentation; NLC: 2D NLC map estimation.

| Exp. | LiDAR | 3D Seg | 3D Ctr | Img | 2D Seg | NLC | Car | Ped. | Cyc. | **mAP** |
|------|-------|--------|--------|-----|--------|-----|-----|------|------|---------|
| (a) | ✓ | | | | | | 87.40 | 59.02 | 79.11 | 75.18 |
| (b) | ✓ | ✓ | | | | | 86.75 | 59.90 | 78.57 | 75.07 |
| (c) | ✓ | | ✓ | | | | 86.56 | 61.06 | 78.58 | 75.40 |
| (d) | ✓ | ✓ | ✓ | | | | 86.71 | 62.23 | 78.68 | 75.88 |
| (e) | ✓ | ✓ | ✓ | ✓ | | | 86.91 | 63.23 | 79.00 | 76.38 |
| (f) | ✓ | ✓ | ✓ | ✓ | ✓ | | 87.10 | 64.52 | 79.94 | 77.18 |
| (g) | ✓ | ✓ | ✓ | ✓ | ✓ | ✓ | 87.18 | 67.52 | 81.21 | 78.64 |

and Table2(b), the mAP is largely boosted from 75.88% to 77.01%. Considering that during the inference stage, these two variants have identical forms of input and network structure, the resulting improvement strongly indicates that the representation ability of the 3D LiDAR branch is indeed strengthened. In other words, the joint optimization of a mature CNN architecture with 2D auxiliary task supervision can guide the point cloud branch to learn more discriminative point features. *Third*, comparing Table2(b) and Table2(c) with Table2(d), we can verify the superiority of bidirectional propagation (78.64%) over the unidirectional schemes (77.01% and 77.47%). If we particularly pay attention to Table2(c) and Table2(d), we can conclude that our newly proposed point-to-pixel feature propagation direction further brings a 1.17% mAP increase based on the previously explored pixel-to-point paradigm.

**Effectiveness of Image Branch.** Comparing Table 3 (d) with Tables 3 (e)–(g), the performance is stably boosted from 75.88% to 78.64%, as the gradual addition of the 2D image branch and two auxiliary tasks including 2D NLC map estimation and semantic segmentation. These results indicate that the additional semantic and geometric features learned from the image modality are indeed effective supplements to the point cloud representation, leading to significantly improved 3D object detection performances.

**Effectiveness of Auxiliary Tasks.** Comparing Table 3 (a) with Tables 3 (b)–(d), the mAP is boosted by 0.70% when incorporating 3D semantic segmentation and 3D center estimation, and can be further improved by 2.76% after introducing 2D semantic segmentation and NLC map estimation (Tables 3 (d)–(g)). However, only integrating image features without 2D auxiliary tasks (comparing results of Tables 3 (d) and (e)) brings limited improvement of 0.5%. This observation shows that the 2D auxiliary tasks, especially the proposed 2D NLC map estimation, do enforce the image branch to learn complementary information for the detector.

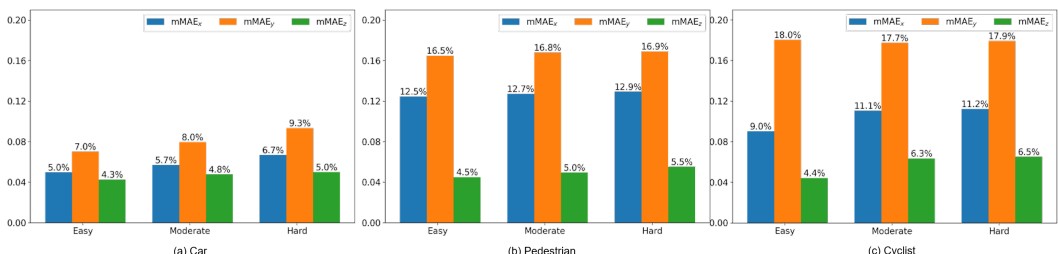

Figure 5: Quantitative results of NLC map estimation on the KITTI val set.

**Performance of NLC Map Estimation.** The NLC map estimation task is introduced to guide the image branch to learn local spatial-aware features. We evaluated the predicted NLC of pixels containing at least one projected point with mean absolute error (MAE) for each object:

$$\text{MAE}_q = \frac{1}{N_{obj}} \sum_i^N \left| q_{\boldsymbol{p}_i}^{NLC} - \hat{q}_{\boldsymbol{p}_i}^{NLC} \right| \cdot \mathbf{1}_{\boldsymbol{p}_i}, q \in \{x, y, z\}, \tag{9}$$

where $N_{obj}$ is the number of LiDAR points inside the ground-truth box of the object, $\mathbf{1}_{\boldsymbol{p}i}$ indicates that the evaluation is only calculated with foreground points inside the box, $q_{\boldsymbol{p}_i}^{NLC}$ and $\hat{q}_{\boldsymbol{p}_i}^{NLC}$ are the

normalized local coordinates of the ground-truth and the prediction at the corresponding pixel for the foreground point. Finally, we obtained the mean value of $\mathrm{MAE}_q$ for all instances, namely $\mathrm{mMAE}_q$. We report the metric over different difficulty levels for three categories on the KITTI validation set in Figure 5. We can observe that, for the car class, the $\mathrm{mMAE}$ error is only 0.0619, i.e., $\pm 6.19$ cm error per meter. For the challenging pedestrian and cyclist categories, the error becomes larger due to the smaller size and non-rigid shape.

## 6 Conclusion

We have introduced a novel cross-modal 3D detector named UPIDet that leverages complementary information from the image domain in two ways. First, we demonstrated that the representational power of the point cloud backbone can be enhanced through the gradients backpropagated from the training loss of the image branch, utilizing a succinct and effective point-to-pixel module. Second, we proposed NLC map estimation as an auxiliary task to facilitate the learning of local spatial-aware representation in addition to semantic features by the image branch. Our approach achieves state-of-the-art results on the KITTI detection benchmark. Furthermore, extensive ablative experiments demonstrate the robustness of UPIDet against sensor noises and its ability to generalize to LiDAR signals with fewer beams. The promising results also indicate the potential of joint training between 3D object detection and other 2D scene understanding tasks. We believe that our novel perspective will inspire further investigations into multi-task multi-modal learning for scene understanding in autonomous driving.

## Acknowledgement

This work was supported in part by the Hong Kong Research Grants Council under Grants CityU 11219422, CityU 11202320, and CityU 11218121, in part by the Hong Kong Innovation and Technology Fund under Grant MHP/117/21, and in part by the CityU Strategic Research Grant CityU 11210523. This work used the computational facilities provided by the Computing Services Centre at the City University of Hong Kong. Besides, we thank the anonymous reviewers for their invaluable feedback that improved our manuscript.

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

# A  Appendix

In this appendix, we provide the details omitted from the manuscript due to space limitation. We organize the appendix as follows.

## A.1  Implementation Details

**Network Architecture.** Fig. A1 illustrates the architectures of the point cloud and image backbone networks. For the encoder of the point cloud branch, we further show the details of multi-scale grouping (MSG) network in Table A1. Following 3DSSD [49], we take key points sampled by the $3^{rd}$ SA layer to generate vote points and estimate the 3D box. Then we feed these 3D boxes and features output by the last FP layer to the refinement stage. Besides, we adopt density-aware RoI grid pooling [11] to encode point density as an additional feature. Note that 3D center estimation [9] aims to learn the relative position of each foreground point to the object center, while the 3D box is estimated based on sub-sampled points. Thus, the auxiliary task of 3D center estimation differs from 3D box estimation and can facilitate learning structure-aware features of objects.

Table A1: Details of set abstraction layers in the point cloud branch. We report the sampling strategy used in the sampling operation, ball radius of group operation, "nquery" that denotes the number of group points, and dimensions of the unit PointNet layer for multi-scale grouping. The features at different scales are concatenated and dimensionally reduced to the specific output channels.

| Layer | Sampling Strategy | Radius | nquery | Feature Dimension | Output Channels |
|---|---|---|---|---|---|
| $1^{st}$ SA | D-FPS | [0.2, 0.4, 0.8] | [32, 32, 64] | [[16, 16, 32], [16, 16, 32], [32, 32, 64]] | 64 |
| $2^{nd}$ SA | D-FPS & S-FPS | [0.4, 0.8, 1.6] | [32, 32, 64] | [[64, 64, 128], [64, 64, 128], [64, 96, 128]] | 128 |
| $3^{rd}$ SA | D-FPS & S-FPS | [1.6, 3.2, 4.8] | [64, 64, 128] | [[128, 128, 256], [128, 196, 256], [128, 256, 256]] | 256 |

**Training Details.** Through the experiments on KITTI dataset, we adopted Adam [13] ($\beta_1$=0.9, $\beta_2$=0.99) to optimize our UPIDet. We initialized the learning rate as 0.003 and updated it with the one-cycle policy [37]. And we trained the model for a total of 80 epochs in an end-to-end manner. In our experiments, the batch size was set to 8, equally distributed on 4 NVIDIA 3090 GPUs. We kept the input image with the original resolution and padded it to the size of $1248 \times 376$, and down-sampled the input point cloud to 16384 points during training and inference. Following the common practice, we set the detection range of the $x$, $y$, and $z$ axis to [0m, 70.4m], [-40m, 40m] and [-3m, 1m], respectively.

**Data Augmentation.** We applied common data augmentation strategies at global and object levels. The global-level augmentation includes random global flipping, global scaling with a random scaling factor between 0.95 and 1.05, and global rotation around the $z$-axis with a random angle in the range of $[-\pi/4, \pi/4]$. Each of the three augmentations was performed with a 50% probability for each sample. The object-level augmentation refers to copying objects from other scenes and pasting them to current scene [47]. In order to perform sampling synchronously on point clouds and images, we utilized the instance masks provided in [29]. Specifically, we pasted both the point clouds and pixels of sampled objects to the point cloud and images of new scenes, respectively.

## A.2  More quantitative Results

**Performance on KITTI Val Set.** We also reported the performance of our UPIDet on all three classes of the KITTI validation set in Table A2, where it can be seen that our UPIDet also achieves the highest mAP of 77.73%, which is obviously higher than the second best method CAT-Det.

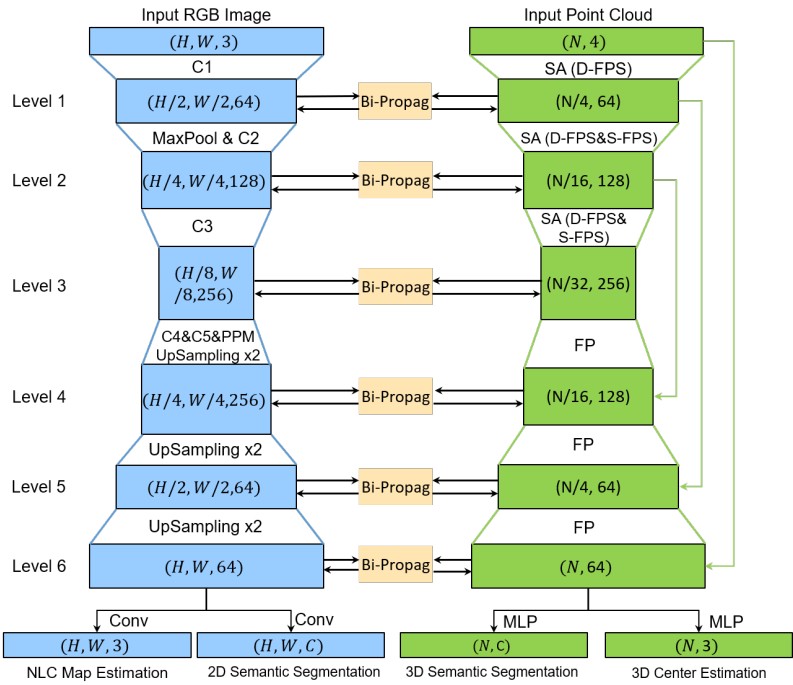

Figure A1: The detailed architecture of 2D and 3D backbones. We adopt ResNet18 as the encoder of the image branch, followed by a decoder with Pyramid Pooling Module (PPM) and several up-sampling blocks. C1, C2, C3, C4, and C5 denote convolutional layers of different stages in ResNet. Extra convolutional layers are deployed after each up-sampling layer. For the point cloud branch, we adopt the PointNet++ structure. SA: set abstraction layer, D-FPS: 3D Euclidean distance-based farthest point sampling, S-FPS: semantic-guided farthest point sampling, FP: feature propagation layer, MLP: shared multi-layer perceptron. Besides, "Bi-Propag" denotes the proposed bidirectional feature propagation between the 2D and 3D backbones.

Table A2: Quantitative comparisons on the KITTI validation set under the evaluation metric of 3D Average Precision (AP) calculated with 11 sampling recall positions. We highlight the best and the second best results in bold and underlined, respectively.

| Method | Modality | 3D Car (IoU=0.7) | | | 3D Ped. (IoU=0.5) | | | 3D Cyc. (IoU=0.5) | | | mAP |
|---|---|---|---|---|---|---|---|---|---|---|---|
| | | Easy | Mod. | Hard | Easy | Mod. | Hard | Easy | Mod. | Hard | |
| PointPillars [15] | LiDAR | 86.46 | 77.28 | 74.65 | 57.75 | 52.29 | 47.90 | 80.05 | 62.68 | 59.70 | 66.53 |
| SECOND [47] | LiDAR | 88.61 | 78.62 | 77.22 | 56.55 | 52.98 | 47.73 | 80.58 | 67.15 | 63.10 | 68.06 |
| 3DSSD [49] | LiDAR | 88.55 | 78.45 | 77.30 | 58.18 | 54.31 | 49.56 | 86.25 | 70.48 | 65.32 | 69.82 |
| PointRCNN [32] | LiDAR | 88.72 | 78.61 | 77.82 | 62.72 | 53.85 | 50.24 | 86.84 | 71.62 | 65.59 | 70.67 |
| PV-RCNN [33] | LiDAR | 89.03 | 83.24 | 78.59 | 63.71 | 57.37 | 52.84 | 86.06 | 69.48 | 64.50 | 71.65 |
| TANet [22] | LiDAR | 88.21 | 77.85 | 75.62 | 70.80 | 63.45 | 58.22 | 85.98 | 64.95 | 60.40 | 71.72 |
| Part-$A^2$ [34] | LiDAR | 89.55 | 79.40 | 78.84 | 65.68 | 60.05 | 55.44 | 85.50 | 69.90 | 65.48 | 72.20 |
| AVOD-FPN [14] | LiDAR+RGB | 84.41 | 74.44 | 68.65 | - | 58.80 | - | - | 49.70 | - | - |
| PointFusion [46] | LiDAR+RGB | 77.92 | 63.00 | 53.27 | 33.36 | 28.04 | 23.38 | 49.34 | 29.42 | 26.98 | 42.75 |
| F-PointNet [28] | LiDAR+RGB | 83.76 | 70.92 | 63.65 | 70.00 | 61.32 | 53.59 | 77.15 | 56.49 | 53.37 | 65.58 |
| CLOCs [26] | LiDAR+RGB | 89.49 | 79.31 | 77.36 | 62.88 | 56.20 | 50.10 | 87.57 | 67.92 | 63.67 | 70.50 |
| EPNet [12] | LiDAR+RGB | 88.76 | 78.65 | 78.32 | 66.74 | 59.29 | 54.82 | 83.88 | 65.50 | 62.70 | 70.96 |
| CAT-Det [54] | LiDAR+RGB | **90.12** | 81.46 | 79.15 | **74.08** | 66.35 | 58.92 | 87.64 | 72.82 | 68.20 | 75.42 |
| UPIDet (Ours) | LiDAR+RGB | 89.73 | **86.40** | **79.31** | 71.77 | **68.49** | **62.52** | **89.24** | **76.91** | **75.18** | **77.73** |

**Performance of Single-Class Detector.** Quite a few methods [6, 60] train models only for car detection. Empirically, the single-class detector performs better in the car class compared with multi-class detectors. Therefore, we also provided performance of UPIDet trained only for the car class, and compared it with several state-of-the-art methods in Table A3.

**Generalization to Asymmetric Backbones.** As shown in Fig. A1, we originally adopted an encoder-decoder network in the LiDAR branch that is architecturally similar to the image backbone. Nevertheless, it is worth clarifying that our approach is not limited to symmetrical structures and

Table A3: Comparison with state-of-the-art methods on the KITTI val set for car 3D detection. All results are reported by the average precision with 0.7 IoU threshold. R11 and R40 denotes AP calculated with 11 and 40 recall sampling recall points, respectively.

| Method | Modal | $\text{AP}_{\text{3D\|R11}}$ (%) | | | $\text{AP}_{\text{3D\|R40}}$ (%) | | |
|---|---|---|---|---|---|---|---|
| | | Easy | Mod. | Hard | Easy | Mod. | Hard |
| Voxel R-CNN [6] | LiDAR | 89.41 | 84.52 | 78.93 | 92.38 | 85.29 | 82.86 |
| PV-RCNN [33] | LiDAR | 89.35 | 83.69 | 78.7 | 92.57 | 84.83 | 82.69 |
| SA-SSD [9] | LiDAR | 90.15 | 79.91 | 78.78 | 93.14 | 84.65 | 81.86 |
| SE-SSD [60] | LiDAR | **90.21** | 85.71 | 79.22 | 93.19 | 86.12 | 83.31 |
| GLENet-VR [57] | LiDAR | 89.93 | 86.46 | 79.19 | **93.51** | 86.10 | 83.60 |
| MV3D [5] | LiDAR+RGB | - | - | - | 71.29 | 62.68 | 56.56 |
| 3D-CVF [51] | LiDAR+RGB | - | - | - | 89.67 | 79.88 | 78.47 |
| UPIDet (Ours) | LiDAR+RGB | 89.72 | **86.52** | **79.34** | 93.03 | **86.46** | **83.73** |

can be generalized to different point-based backbones. Here, we replaced the 3D branch of the original framework with an efficient single-stage detector—SASA [2], using a backbone only with the encoder in the LiDAR branch, which is asymmetric with the encoder-decoder structure of the image backbone. Accordingly, the proposed bidirectional propagation is only performed between the 3D backbone and the encoder of the 2D image backbone. The experimental results are shown in Table A4. We can observe that the proposed method works well even when the two backbones are asymmetric, which demonstrates the satisfactory generalization ability of our method for different LiDAR backbones.

Table A4: Bidirectional propagation is also effective when the 2D and 3D backbones are asymmetric. Here we adopt a single-stage detector [2] whose backbone includes only an encoder, which is asymmetric with the encoder-decoder network in the image branch.

| Method | 3D Car (IoU=0.7) | | | 3D Ped. (IoU=0.5) | | | 3D Cyc. (IoU=0.5) | | | mAP |
|---|---|---|---|---|---|---|---|---|---|---|
| | Easy | Mod. | Hard | Easy | Mod. | Hard | Easy | Mod. | Hard | |
| SASA | 92.17 | 84.90 | 82.57 | 66.75 | 61.40 | 56.00 | 89.91 | 74.05 | 69.41 | 75.24 |
| Ours (SASA) | 92.11 | 85.67 | 82.99 | 70.52 | 63.38 | 58.16 | 91.58 | 74.81 | 70.22 | 76.61 |

Table A5: 3D detection results on the Waymo Open Dataset validation set. "-" denotes that the results are not reported in their papers.

| Method | Vehicle L1 | | Vehicle L2 | | Pedestrian L1 | | Pedestrian L2 | | Cyclist L1 | | Cyclist L2 | |
|---|---|---|---|---|---|---|---|---|---|---|---|---|
| | mAP | mAPH | mAP | mAPH | mAP | mAPH | mAP | mAPH | mAP | mAPH | mAP | mAPH |
| PointAugmenting | 67.41 | - | 62.7 | - | 75.42 | - | 70.55 | - | 76.29 | - | 74.41 | - |
| TransFusion | - | - | - | 65.14 | - | - | - | 64.00 | - | - | - | 67.40 |
| Ours | 78.36 | 77.91 | 69.45 | 69.04 | 76.32 | 71.67 | 65.93 | 61.81 | 79.64 | 78.55 | 76.36 | 75.26 |

## A.3  Results on Waymo Open Dataset

The Waymo Open Dataset [38] is a large-scale dataset for 3D object detection. It contains 798 sequences (15836 frames) for training, and 202 sequences (40077 frames) for validation. According to the number of points inside the object and the difficulty of annotation, the objects are further divided into two difficulty levels: LEVEL_1 and LEVEL_2. Following common practice, we adopted the metrics of mean Average Precision (mAP) and mean Average Precision weighted by heading accuracy (mAPH), and reported the performance on both LEVEL_1 and LEVEL_2. We set the detection range to [-75.2m, 75.2m] for $x$ and $y$ axis, and [-2m, 4m] for $z$ axis. Following [41] and [1], the training on Waymo dataset consists of two stages to allow flexible augmentations. First, we only trained the LiDAR branch without image inputs and bidirectional propagation for six epochs. We enabled the copy-and-paste augmentation in this stage. Then, we trained the whole pipeline for another 30 epochs, during which the copy-and-paste is disabled. Note that the image semantic segmentation head is disabled, since ground-truth segmentation maps are not provided [38].

As shown in Table A5, our method achieves substantial improvement compared with previous state-of-the-arts. Particularly, unlike existing approaches including PointAugmenting [41] and TransFusion [1] where the camera backbone is pre-trained on other datasets and then frozen, we trained the entire

pipeline in an end-to-end manner. It can be seen that even without the 2D segmentation auxiliary task, our method still achieves higher accuracy under all scenarios except "Ped L2", demonstrating its advantage.

## A.4 More Ablation Studies

Table A6: Effect of the semantic-guided SA layer. Compared with the single-modal baseline, UPIDet can better exploit image semantics and preserve more foreground points during downsampling.

| Method | Single-Modal | | UPIDet (Ours) | | *Improvement* | |
|---|---|---|---|---|---|---|
| SA Layer | FG Rate | Instance Recall | FG Rate | Instance Recall | FG Rate | Instance Recall |
| Level-2 | 15.87 | 97.92 | 20.70 | 98.23 | *+4.83* | *+0.31* |
| Level-3 | 29.73 | 97.35 | 38.03 | 97.82 | *+8.29* | *+0.47* |

Table A7: Comparison between our multi-task training methods and the single-modal 2D semantic segmentation baseline (PSPNet). The results show that point features effectively improve the segmentation performance on pedestrian and cyclist classes.

| Method | Car | Pes. | Cyc. | mIoU |
|---|---|---|---|---|
| PSPNet [59] | 77.49 | 30.45 | 23.83 | 43.92 |
| UPIDet (Ours) | 78.45 | 36.15 | 30.42 | 48.34 |

**Effect of Semantic-guided Point Sampling.** When performing downsampling in the SA layers of the point cloud branch, we adopted S-FPS [2] to explicitly preserve as many foreground points as possible. We report the percentage of sampled foreground points and instance recall (i.e., the ratio of instances that have at least one point) in Table A6, where it can be seen that exploiting supplementary semantic features from images leads to substantial improvement of the ratio of sampled foreground points and better instance recall during S-FPS.

**Influence on 2D Semantic Segmentation.** We also aimed to demonstrate that the 2D-3D joint learning paradigm benefits not only the 3D object detection task but also the 2D semantic segmentation task. As shown in Table A7, the deep interaction between different modalities yields an improvement of 4.42% mIoU. The point features can naturally complement RGB image features by providing 3D geometry and semantics, which are robust to illumination changes and help distinguish different classes of objects, for 2D visual information. The results suggest the potential of joint training between 3D object detection and more 2D scene understanding tasks in autonomous driving.

**Conditional Analysis.** To better figure out where the improvement comes from when using additional image features, we compared UPIDet with the single-modal detector on different occlusion levels and distant ranges. The results shown in Table A8 and Table A9 include separate APs for objects belonging to different occlusion levels and APs for moderate class in different distance ranges. For car detection, our UPIDet achieves more accuracy gains for long-distance and highly occluded objects, which suffer from the sparsity of observed LiDAR points. The cyclist and pedestrian are much more difficult categories on account of small sizes, non-rigid structures, and fewer training samples. For these two categories, UPIDet still brings consistent and significant improvements on different levels even in extremely difficult cases.

Table A8: Performance breakdown over different occlusion levels. As defined by the official website of KITTI, occlusion levels 0, 1, and 2 correspond to fully-visible samples, partly-occluded samples, and samples that are difficult to see, respectively.

| Class | Car | | | Pedestrian | | | Cyclist | | |
|---|---|---|---|---|---|---|---|---|---|
| Occlusion | Level-0 | Level-1 | Level-2 | Level-0 | Level-1 | Level-2 | Level-0 | Level-1 | Level-2 |
| Single-Modal | 91.98 | 77.18 | 55.41 | 67.44 | 26.76 | 6.10 | 91.23 | 24.66 | 1.74 |
| UPIDet (Ours) | 92.26 | 77.44 | 58.39 | 74.00 | 35.13 | 7.99 | 92.89 | 30.02 | 2.53 |
| *Improvement* | *+0.28* | *+0.26* | *+2.97* | *+6.56* | *+8.38* | *+1.89* | *+1.66* | *+5.36* | *+0.79* |

**Generalization to Sparse LiDAR Signals.** We also compared our UPIDet with the single-modal baseline on LiDAR point clouds with various sparsity. In practice, following Pseudo-LiDAR++ [52], we simulated the 32-beam, 16-beam, and 8-beam LiDAR signals by selecting LiDAR points whose

Table A9: Performance breakdown over different distances.

| Class | Car | | | Pedestrian | | | Cyclist | | |
|---|---|---|---|---|---|---|---|---|---|
| Distance | 0-20m | 20-40m | 40m-Inf | 0-20m | 20-40m | 40m-Inf | 0-20m | 20-40m | 40m-Inf |
| Single-Modal | 96.28 | 85.21 | 43.91 | 71.28 | 38.42 | 1.63 | 93.61 | 61.56 | 34.48 |
| UPIDet (Ours) | 96.36 | 86.48 | 49.88 | 76.56 | 45.72 | 2.46 | 94.12 | 67.27 | 39.10 |
| *Improvement* | +0.07 | +1.27 | +5.97 | +5.28 | +7.30 | +0.83 | +0.51 | +5.71 | +4.62 |

elevation angles fall within specific intervals. As shown in Table A10, the proposed UPIDet outperforms the single-modal baseline under all settings. The consistent improvements suggest our method can generalize to sparser signals. Besides, the proposed UPIDet significantly performs better than the baseline in the setting of LiDAR signals with fewer beams, demonstrating the effectiveness of our method in exploiting the supplementary information in the image domain.

Table A10: Comparison with single-modal baselines under LiDAR signals with different beams, where we report $AP_{3D|R40}$ on the KITTI validation set.

| LiDAR Beams | Modal | Car | Ped. | Cyc. | **mAP** |
|---|---|---|---|---|---|
| 64 | LiDAR | 86.71 | 62.23 | 78.68 | 75.87 |
| | LiDAR + RGB | 87.18 | 67.52 | 81.21 | 78.64 |
| | *Improvement* | +0.47 | +5.29 | +2.53 | +2.76 |
| 32 | LiDAR | 83.49 | 57.83 | 70.82 | 70.71 |
| | LiDAR + RGB | 84.47 | 62.56 | 73.93 | 73.65 |
| | *Improvement* | +0.98 | +4.73 | +3.11 | +2.94 |
| 16 | LiDAR | 79.80 | 53.84 | 60.51 | 64.71 |
| | LiDAR + RGB | 80.78 | 59.44 | 67.35 | 69.19 |
| | *Improvement* | +0.99 | +5.60 | +6.84 | +4.48 |
| 8 | LiDAR | 64.42 | 22.57 | 43.19 | 43.39 |
| | LiDAR + RGB | 67.09 | 31.03 | 47.97 | 48.70 |
| | *Improvement* | +2.66 | +8.46 | +4.78 | +5.30 |

**Robustness against Input Corruption.** We also conducted extensive experiments to verify the robustness of our UPIDet to sensor perturbation. Specifically, we added Gaussian noises to the reflectance value of points or RGB images. Fig. A2 shows that the mAP value of our cross-modal UPIDet is consistently higher than that of the single-modal baseline and decreases slower with the LiDAR noise level increasing. Particularly, as listed in Table A11, when the variance of the LiDAR noise is set to 0.15, the perturbation affects our cross-modal UPIDet much less than the single-modal detector. Besides, even applying the corruption to both LiDAR input and RGB images, the mAP value of our UPIDet only drops by 2.49%.

Table A11: Performances (mAPs) of the single-modal baseline and our UPIDet on the KITTI val set under input corruptions of simulated LiDAR and image noise sampled from the Gaussian distribution. Note that the image noise is only applicable to multi-modal detectors.

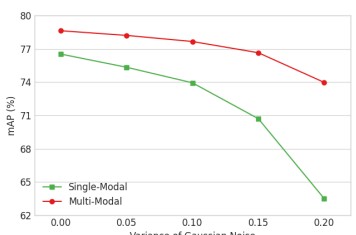

Figure A2: Comparisons of noise robustness between the single-modal baseline and our UPIDet.

| Corruptions Type | Modal | Car | Ped. | Cyc. | **mAP** |
|---|---|---|---|---|---|
| No Corruption | LiDAR | 86.71 | 62.23 | 78.68 | 75.87 |
| LiDAR Noise | LiDAR | 84.37 | 49.19 | 77.27 | 70.28 |
| No Corruption | LiDAR + RGB | 87.18 | 67.52 | 81.21 | 78.64 |
| LiDAR Noise | LiDAR + RGB | 86.82 | 65.61 | 77.63 | 76.69 |
| Image Noise | LiDAR + RGB | 87.14 | 66.26 | 77.97 | 77.12 |
| LiDAR + Image Noise | LiDAR + RGB | 86.60 | 65.42 | 76.44 | 76.15 |

**Effectiveness of Multi-stage Interaction.** As mentioned before, both 2D and 3D backbones adopt an encoder-decoder structure, and we perform bidirectional feature propagation at both downsampling and upsampling stages. Here, we conducted experiments to verify the superiority of the multi-stage interaction over single-stage interaction. As shown in Table A12, only performing the bidirectional feature propagation in the encoder (i.e., Table A12 (b)) or the decoder (i.e., Table A12 (c)) leads to worse performance than that of performing the module in both stages (i.e., Table A12 (d)).

Table A12: Ablative experiments on the multi-stage manner of bidirectional propagation, where SA and FP denote applying bidirectional propagation at downsampling (in the encoder) and upsampling (in the decoder) stages of the point cloud branch, respectively.

|  | Stage | | 3D Car (IoU=0.7) | | | 3D Ped. (IoU=0.5) | | | 3D Cyc. (IoU=0.5) | | | mAP |
|  | SA | FP | Easy | Mod. | Hard | Easy | Mod. | Hard | Easy | Mod. | Hard | |
|---|---|---|---|---|---|---|---|---|---|---|---|---|
| (a) | - | - | 91.92 | 85.22 | 82.98 | 68.82 | 61.47 | 56.39 | 91.93 | 74.56 | 69.58 | 75.88 |
| (b) | ✓ | - | 92.67 | 85.86 | 83.40 | 70.94 | 65.29 | 60.35 | 93.60 | 75.60 | 71.04 | 77.64 |
| (c) | - | ✓ | 92.19 | 85.44 | 83.27 | 69.02 | 63.41 | 58.47 | 92.85 | 76.39 | 71.91 | 76.99 |
| (d) | ✓ | ✓ | 92.63 | 85.77 | 83.13 | 72.68 | 67.64 | 62.25 | 94.39 | 77.77 | 71.47 | 78.64 |

## A.5 Efficiency Analysis

We also compared the inference speed and number of parameters of the proposed UPIDet with state-of-the-art cross-modal approaches in Table A13. Our UPIDet has about the same number of parameters as CAT-Det [54], but a much higher inference speed at 9.52 frames per second on a single GeForce RTX 2080 Ti GPU. In general, our UPIDet is inevitably slower than some single-modal detectors, but it achieves a good trade-off between speed and accuracy among cross-modal approaches.

Table A13: Comparison of the number of network parameters, inference speed, and detection accuracy of different multi-modal methods on the KITTI test set.

| Method | Params (M) | Frames per second | mAP (%) |
|---|---|---|---|
| AVOD-FPN [14] | 38.07 | 10.00 | 56.84 |
| F-PointNet [28] | 12.45 | 6.25 | 57.86 |
| EPNet [12] | 16.23 | 5.88 | - |
| CAT-Det [54] | 23.21 | 3.33 | 67.05 |
| UPIDet (Ours) | 24.98 | 9.52 | 70.13 |

## A.6 Visual Results of 3D Object Detection

In Figure A3, we present the qualitative comparison of detection results between the single-modal baseline and our UPIDet. We can observe that the proposed UPIDet shows better localization capability than the single-modal baseline in challenging cases. Besides, we also show qualitative results of UPIDet on the KITTI test split in Figure A4. We can clearly observe that our UPIDet performs well in challenging cases, such as pedestrians and cyclists (with small sizes) and highly-occluded cars.

## A.7 Visual Results of 2D Semantic Segmentation

Several examples are shown in Figure A5. For distant cars in the first and the second row as well as the pedestrian in the sixth row, the size of objects is small and PSPNet tends to treat them as background, while our UPIDet is able to correct such errors. In the third row, our UPIDet finds the dim cyclist missed by PSPNet. Our UPIDet also performs better for the highly occluded objects as shown in the fourth and the fifth lines. This observation shows the 3D feature representations extracted from point clouds can boost 2D semantic segmentation, since the image-based method is sensitive to illumination and can hardly handle corner cases with only single-modal inputs.

## A.8 Details on Official KITTI Test Leaderboard

We submitted the results of our UPIDet to the official KITTI website, and it ranks $\mathbf{1}^{st}$ on the 3D object detection benchmark for the cyclist class. Figure A6 shows the screenshot of the leaderboard. Figure A7 illustrates the precision-recall curves along with AP scores on different categories of the KITTI test set. The samples of the KITTI test set are quite different from those of training/validation set in terms of scenes and camera parameters, so the impressive performance of our UPIDet on the test set demonstrates it also achieves good generalization.

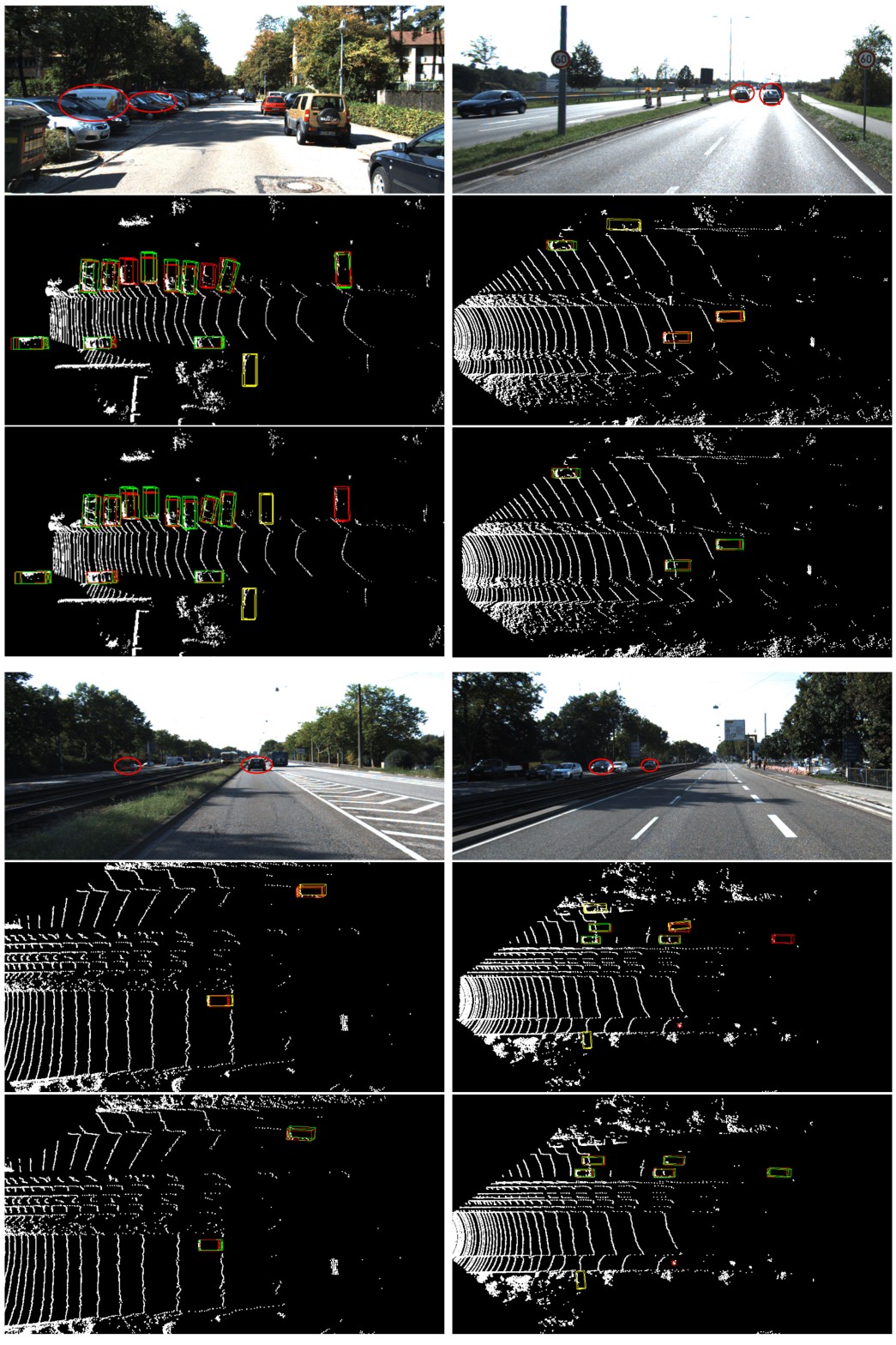

Figure A3: Qualitative comparison between single-modal baseline and our multi-modal UPIDet. For each comparison, from top to bottom, we have the image, detection results of single-modal baseline, and detection results of UPIDet. We use red, green, and yellow to denote the ground-truth, true positive and false positive bounding boxes, respectively. We highlight some objects in images with red circles, which are detected by UPIDet but missed by the single-modal method.

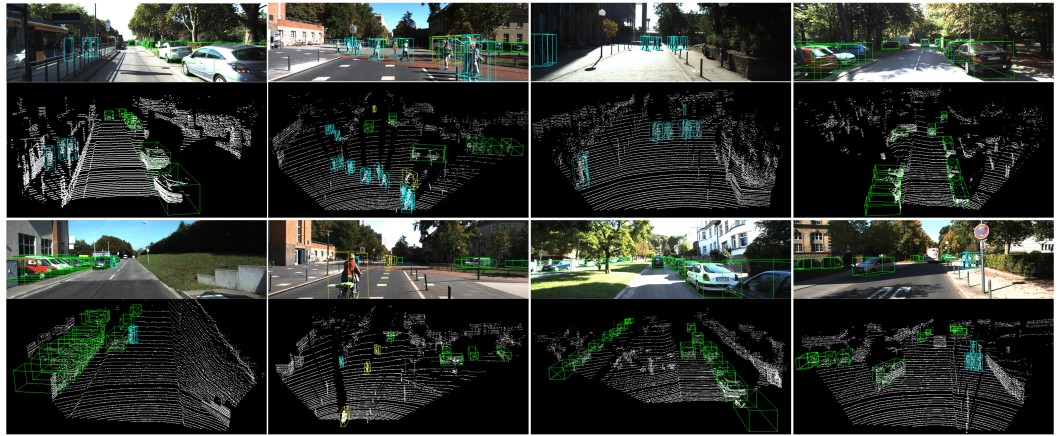

Figure A4: Extra qualitative results of UPIDet on the KITTI test set. The predicted bounding boxes of car, pedestrian, and cyclist are visualized in green, cyan, and yellow, respectively. We also show the corresponding projection of boxes on images. Best viewed in color and zoom in for more details.

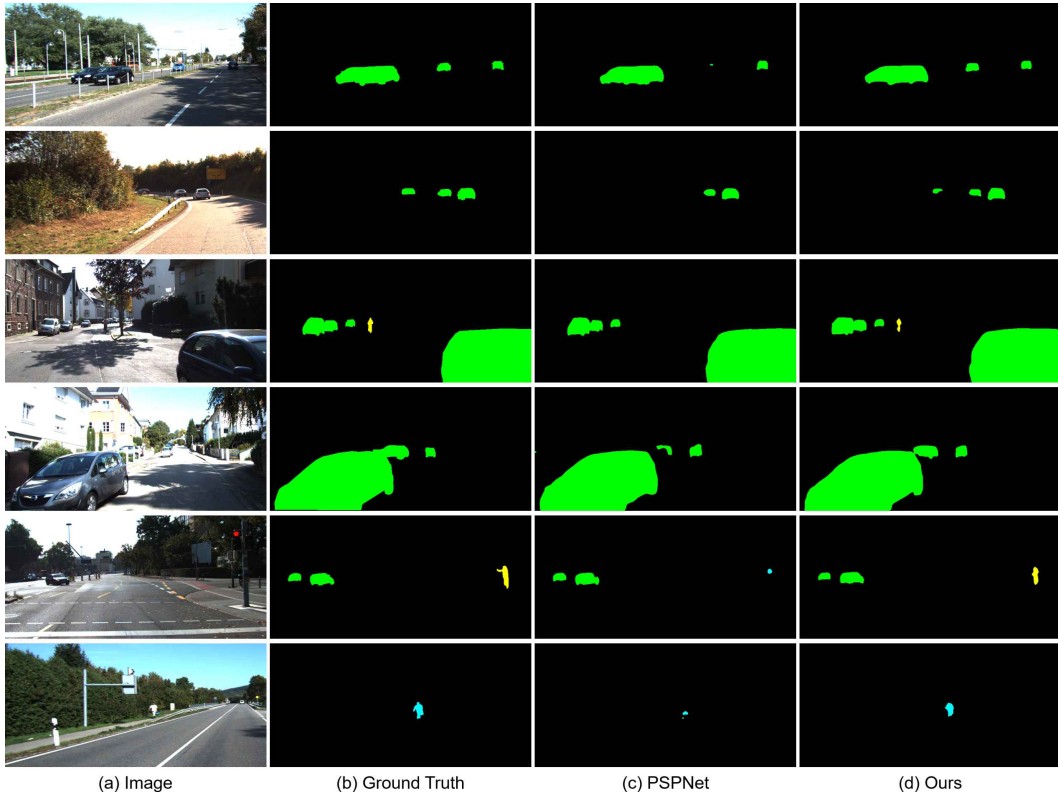

| (a) Image | (b) Ground Truth | (c) PSPNet | (d) Ours |

Figure A5: Visual results of 2D semantic segmentation on the KITTI val set. The prediction boxes are shown in green for car, cyan for pedestrian, and yellow for cyclist. Best viewed in color. Compared with PSPNet, our cross-modal UPIDet produces more accurate and detailed results.

| | Method | Setting | Code | Moderate | Easy | Hard | Runtime | Environment |
|---|---|---|---|---|---|---|---|---|
| 1 | BiProDet | | | 74.32 % | 86.74 % | 67.45 % | 0.1 s | GPU @ 2.5 Ghz (Python + C/C++) |
| 2 | TED | | | 74.12 % | 88.82 % | 66.84 % | 0.1 s | 1 core @ 2.5 Ghz (C/C++) |
| 3 | CasA++ | | code | 73.79 % | 87.76 % | 66.84 % | 0.1 s | 1 core @ 2.5 Ghz (C/C++) |
| 4 | CasA | | code | 73.47 % | 87.91 % | 66.17 % | 0.1 s | 1 core @ 2.5 Ghz (C/C++) |
| 5 | SGNet | | | 70.40 % | 86.75 % | 62.73 % | 0.09 s | GPU @ 2.5 Ghz (Python) |
| 6 | HMFI | | code | 70.37 % | 84.02 % | 62.57 % | 0.1 s | 1 core @ 2.5 Ghz (C/C++) |
| 7 | CAD | | | 69.94 % | 84.68 % | 62.21 % | 0.1 s | GPU @ 2.5 Ghz (Python + C/C++) |
| 8 | SARFE | | | 69.67 % | 84.88 % | 62.26 % | 0.03 s | 1 core @ 2.5 Ghz (C/C++) |
| 9 | EQ-PVRCNN | | code | 69.10 % | 85.41 % | 62.30 % | 0.2 s | GPU @ 2.5 Ghz (Python + C/C++) |
| 10 | VoCo | | | 69.00 % | 82.74 % | 62.46 % | 0.1 s | 1 core @ 2.5 Ghz (Python + C/C++) |

Figure A6: Screenshot of the KITTI 3D object detection benchmark for cyclist class on August 15th, 2022. Note: When we submit the results, the temporary name is BiProDet.

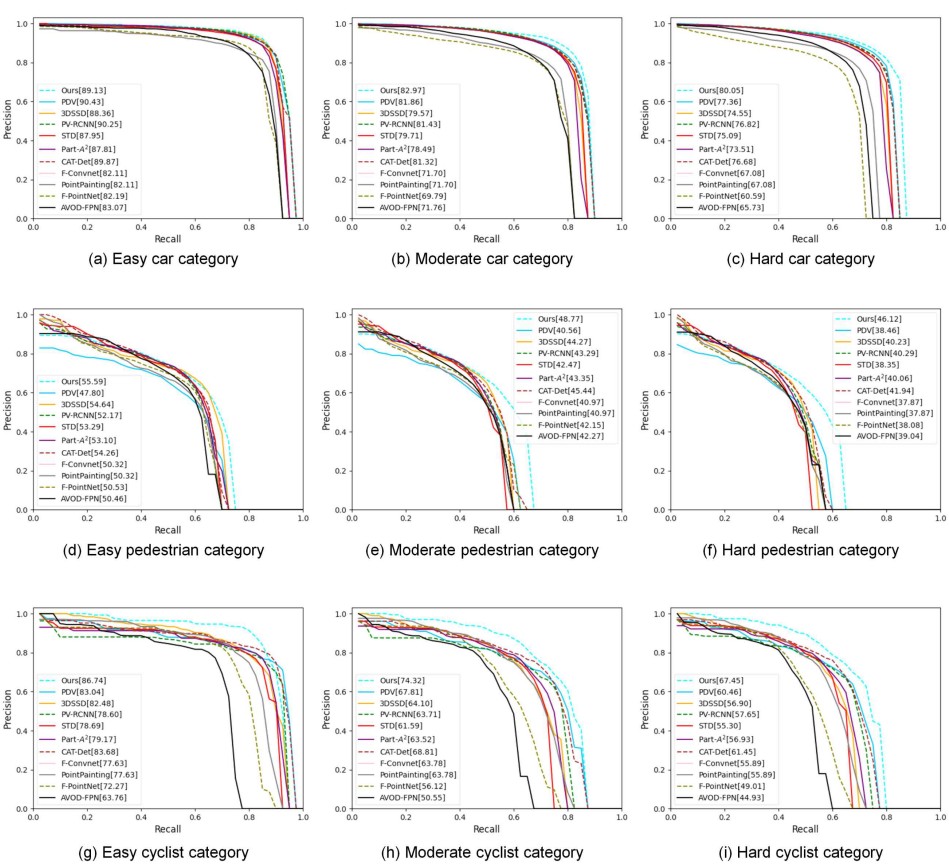

Figure A7: Precision-recall curves of different methods on the KITTI 3D object detection test set on Aug. 15$^{th}$, 2022. We also report APs in different categories for each method.

