# OpenReview forum: "Unleash the Potential of Image Branch for Cross-modal 3D Object Detection"
_NeurIPS.cc/2023/Conference — NeurIPS 2023 poster_

### Official Review · Reviewer_DmuL · 2023-07-04

**Soundness:** 2 fair
**Presentation:** 2 fair
**Contribution:** 2 fair
**Rating:** 5
**Confidence:** 4

**Summary:**

This paper proposes a novel cross-model 3D detector BiProDet, which leverages the information from image domain in two ways. First, it proposes point-to-pixel bidirectional propagation strategy to boost the representation ability of the point cloud backbone. Second, it introduces NLC map estimation as an auxiliary task for training to learn local spatial-aware features of image branch to supply sparse point clouds.  The BiProDet exhibits consistent and more significant improvements on the "moderate" and "hard" levels, where objects are distant or highly occluded with sparse points and ranked 1st on the KITTI 3D detection benchmark for the cyclist class.
In my opinion, this paper discusses of the reinforcement mechanism of image branch for 3D detector deeply, and a bidirectional propagation is designed to enhance the performance of the backbone, which is very innovative. And a wealth of ablation experiments has been done to prove the effectiveness of the algorithm proposed in this paper.

**Strengths:**

The paper has several strengths in terms of originality, quality, clarity, and significance.

1. It introduces a 2D auxiliary task called normalized local coordinate (NLC) map estimation to improve the performance of the cross-modal detector by providing the relative position of each pixel inside the object.

2. The paper proposes a novel point-to-pixel feature propagation mechanism that allows 3D geometric features from LiDAR point clouds to enhance the representation ability of 2D image learning.

3. It provides a analysis of the performance bottlenecks of single-modal
   detectors.

**Weaknesses:**

1. The motivation is weak. As the proposed method point-to-pixel is just a fusion strategy in cross-modal modeling, which also been explored in many works such as BEVFusion[1]. So it is relatively weak to serve as the motivation of the paper.

2. The proposed specific 2D auxiliary tasks have been explored in TiG-BEV [2]. Althogh the format of the 2D tasks is not the same, the core of the methods is similar.

[1] https://github.com/mit-han-lab/bevfusion

[2] TiG-BEV: Multi-view BEV 3D Object Detection via Target Inner-Geometry Learning

**Questions:**

None

---

> ### Author Rebuttal · Authors · 2023-08-09
>
> ### Response to Reviewer DmuL
>
> We sincerely appreciate the reviewer for your time and effort in reviewing our paper. In the following, we will comprehensively address your concerns.
>
> ### **Comment 1:** *The motivation is weak. As the proposed method point-to-pixel is just a fusion strategy in cross-modal modeling, which also been explored in many works such as BEVFusion[1]. So it is relatively weak to serve as the motivation of the paper.*
>
> **Response:** We agree that the bidirectional fusion mechanisms have been explored before. However, the motivation of our work is **not** to obtain stronger fused feature representation with trivial structural innovation of the point-to-pixel module like previous works. Instead, recent research [R1] shows that the representation capability of point-based backbone networks for 3D point clouds is still relatively **insufficient** due to the irregular characteristic, which limits the performance of 3D detectors. Therefore, we aim to directly **boost the representation capability of the point cloud backbone** network by back-propagating **gradients** from the training objectives of the image branch through the proposed point-to-pixel module. We experimentally demonstrated our claim in Table 2 of our manuscript, where the mAP largely boosted from 75.88% (Table2(a)) to 77.01% (Table2(b)), **precisely** indicating that the point-to-pixel propagation indeed strengthens the representation ability of the 3D LiDAR branch. To the best of our knowledge, it has never been identified previously that **2D** auxiliary tasks can be used to improve the representation ability of the **3D** backbone. Besides, **Reviewer gAXu** also acknowledged such a manner is non-trivial and interesting.
>
> [R1] Zhang et al., PointMCD: Boosting Deep Point Cloud Encoders via Multi-view Cross-modal Distillation for 3D Shape Recognition, TMM'23.
>
>
> ### **Comment 2:** *The proposed specific 2D auxiliary tasks have been explored in TiG-BEV [2]. Although the format of the 2D tasks is not the same, the core of the methods is similar.*
>
> **Response:** We agree with the reviewer that the adopted 2D supervision tasks are similar. The camera-based TiG-BEV utilizes inner-depth supervision to enhance the understanding of object-wise spatial structures. By contrast, our cross-modal method incorporates the image branch to learn local spatial-aware features, complementing the information the sparse point clouds provide. It is worth noting that we have made our article and source code publicly available **prior** to the release of TiG-BEV.
>
> It would be better to **holistically** understand the two key components, i.e., the NLC map estimation and point-to-pixel module,  rather than in isolation. On one hand, we can learn extra information from the image branch with auxiliary tasks. On the other hand, we demonstrate that the gradients back-propagated from the image branch through the point-to-pixel module can boost the representation ability of the point cloud backbone. The suitable 2D tasks and the point-to-pixel module are **intricately linked**, and it is only by combining them that the greatest performance improvement can be achieved.

---

> ### Comment · Area_Chair_vkAG · 2023-08-16
> **Rebuttal?**
>
> Dear reviewer,
>
> Can I ask you to please see if the rebuttal addresses your concerns?

---

> > ### Comment · Reviewer_DmuL · 2023-08-17
> >
> > The rebuttal solves my concern on the novelty part. For the comment 1, using gradient from image branch to help the lidar branch is interesting. For the comment 2, to support your claim that "The suitable 2D tasks and the point-to-pixel module are intricately linked", is there any ablation on considering "the NLC map estimation and point-to-pixel module" in isolation or as a whole?

---

> > > ### Author Response · Authors · 2023-08-18
> > > **Thanks for your valuable feedback and your recognition！**
> > >
> > > Thank you for providing your valuable feedback. We are delighted to learn that our rebuttal effectively addresses your concerns regarding the novelty issue. Your recognition of our interesting idea is greatly appreciated.
> > >
> > >
> > > In the manuscript, we have presented the results of the corresponding ablation studies on the relationship between NLC map estimation and the point-to-pixel module. These results can be found in **Table 2** and **Table 3**. To facilitate your understanding, we will now summarize the findings to support our claim about the intricate relationship between NLC map estimation and the point-to-pixel module.
> > >
> > >
> > > As depicted in the table below, it becomes evident that the introduction of either the point-to-pixel module or the NLC map supervision individually results in mAP enhancements of 0.43% and 0.72%, respectively. However, the combination of both elements yields the most significant improvement, leading to an impressive increase of 1.89% in mAP.
> > >
> > > | Exp. | Point2Pixel | 2D NLC | Veh. | Ped.   | Cyc.   | Mean   |
> > > | :--: | :---------: | :---------: | :----: | :----: | :----: | :----: |
> > > | (a)  | &#10006; | &#10006; | 86\.99 | 63\.78 | 79\.49 | 76\.75 |
> > > | (b)  | &#10004; | &#10006; | 87\.10 | 64\.52 | 79\.94 | 77\.18 |
> > > | (c)  | &#10006; | &#10004; | 86\.94 | 64\.85 | 80\.62 | 77\.47 |
> > > | (d)  | &#10004; | &#10004; | 87\.18 | 67\.52 | 81\.21 | 78\.64 |
> > >
> > > We hope that this summary provides further clarity on the relationship between NLC map estimation and the point-to-pixel module. In the final version, we will highlight this point. Once again, we sincerely appreciate your valuable feedback and support.

---

> > > > ### Comment · Reviewer_DmuL · 2023-08-21
> > > >
> > > > Thank you for the feedback. I may raise my rating to 5.

---

> > > > > ### Author Response · Authors · 2023-08-21
> > > > > **Thank you for the feedback**
> > > > >
> > > > > Dear Reviewer DmuL,
> > > > >
> > > > > We sincerely thank you for reconsidering our work in light of our rebuttal. We deeply appreciate your favorable recommendation and the acknowledgment of our efforts. Your valuable feedback has been instrumental in refining our work, and we're truly grateful for the opportunity to address your concerns.
> > > > >
> > > > > Warm regards,
> > > > >
> > > > > Authors

---

### Official Review · Reviewer_s1rE · 2023-07-06

**Soundness:** 3 good
**Presentation:** 3 good
**Contribution:** 2 fair
**Rating:** 4
**Confidence:** 4

**Summary:**

This paper studies 3D object detection with multi-modal inputs (image and point cloud). This study uses a two-stage 3D object detection pipeline and proposes two approaches for further performance improvements. Firstly, the authors propose a bidirectional feature module to fuse point cloud and image features. Secondly, the authors propose several 2D and 3D auxiliary tasks to improve representation learning. The experiments validated the proposed approaches on KITTI and Waymo Open Dataset.

**Strengths:**

1. the ablation study is well-designed and extensive to validate the effectiveness of the proposed NLC map representation and fusion network.
2. the idea of using NLC map is well-motivated.
3. the performance on KITTI and Waymo Open Dataset is competitive in several metrics.

**Weaknesses:**

1. the technical novelty is somewhat limited. Bidirectional feature fusion is not new, for example, see "FFB6D: A Full Flow Bidirectional Fusion Network for 6D Pose Estimation". The proposed fusion approach is similar to this paper, which is not cited. In addition, using auxiliary loss to improve representation learning is also not very novel.
2. Auxiliary tasks require extra labels. Do other approaches in Table 1 use semantic masks as supervision?
3. Compared with [45], the performance in the EASY category is inferior.
4. Previous fusion-base approaches such as [1] provided evaluations on the Nuscenes dataset yet this paper does not.


**Questions:**

1. Why the performance on the EASY category of KITTI is inferior compared with [45]? Can the authors further improve it?
2. Can the authors provide quantitative comparisons with [1] on the Nuscenes dataset?
3. have the authors attempted other fusion approaches instead of the proposed one? What will be the performance difference?

**Limitations:**

 The authors have not addressed the limitations.

---

> ### Author Rebuttal · Authors · 2023-08-09
>
> ### Response to Reviewer s1rE
>
> We sincerely appreciate you raising insightful points that helped improve our work. The comments have helped us better articulate the key contributions and value of our work.
>
> ### **Comment 1:** *The technical novelty is somewhat limited. Bidirectional feature fusion is not new.*
>
> **Response:** We agree that bidirectional fusion techniques have been explored in prior works. However, the main contribution is **not** proposing a structurally new fusion module, e.g., improving the bidirectional fusion module with an attention mechanism. Instead, we aim to **directly boost the representation capability of the point cloud backbone** network by back-propagating **gradients** from the training objectives of image branch through the proposed point-to-pixel module, due to the relatively weak representation capability of current point-based backbones for processing irregular 3D point clouds (see Lines 39-46 of the manuscript). We experimentally demonstrated our claim in Table 2 of our manuscript, where the mAP largely boosted from 75.88% (Table2(a)) to 77.01% (Table2(b)), **precisely** indicating that the point-to-pixel propagation indeed strengthens the representation ability of the 3D LiDAR branch. To the best of our knowledge, it has never been identified previously that 2D auxiliary tasks can improve the representation ability of 3D backbone. Besides, **Reviewer gAXu** also acknowledged such a manner is non-trivial and interesting.
>
> ### **Comment 2:** *Using auxiliary loss to improve representation learning is not very novel.*
>
> **Response:** Previous works have used 3D and 2D auxiliary losses to improve the representation ability of 3D and 2D backbones respectively. And our key finding is that the gradients back-propagated from the training objectives of the image branch can boost the representation ability of the point cloud backbone (see Table 2 of our manuscript). To the best of our knowledge, it has **never** been identified previously that **2D** auxiliary tasks could so effectively improve the representation power of **3D** backbone.
>
> ### **Comment 3:** *Auxiliary tasks require extra labels. Do other approaches in Table 1 use semantic masks as supervision?*
>
> **Response:** Thanks for raising a fair point. Among the several methods listed in Table 1, MMF used extra depth completion supervision and PointPainting used extra semantic segmentation labels. Besides, it is **common** to use extra image segmentation labels in cross-modal 3D detectors, such as EPNet and MVX-Net.
>
> ### **Comment 4:** *Compared with [45], the performance in the EASY category is inferior. Can the authors further improve it?*
>
> **Response:** Thank you for the detailed review. Our method particularly benefits challenging cases with **sparser** point clouds, which are more likely to occur for distant or occluded objects in the moderate and hard categories. Specifically, in this work, we learn local spatial-aware features in the image branch under the supervision of 2D NLC map supervision, which serves as a complement to the sparse point clouds. While [45] presents a novel embedding-querying paradigm and achieves higher APs in the easy category, our overall performance **significantly** surpasses it. Following your suggestion, we will explore ways to improve the AP for easy cases by incorporating stronger image backbones and leveraging the recent advances in the field of 3D object detection.
>
> ### **Comment 5:** *Previous fusion-base approaches such as [1] provided evaluations on the Nuscenes dataset yet this paper does not.*
>
> **Response:** We appreciate the suggestion for including experiments on the NuScenes dataset. To be honest, it is very difficult to achieve competitive performance on the large-scale Nuscenes dataset in such a short time with limited GPU resources. Especially, the baseline **point-based** detector we used performs **poorly** on NuScenes or Waymo, which increases the experiment workload. Actually, when preparing this work, we have made great efforts to improve the performance of point-based detectors on the Waymo dataset to be **comparable** to the BEV-based detectors. Despite the missing experiments on NuScenes, we posit our method sheds light on point-based methods by offering new insights and an effective framework. Also, we believe the strong results on KITTI and Waymo sufficiently demonstrate the effectiveness of our proposed method and validate our motivation.
>
> ### **Comment 6:** *Have the authors attempted other fusion approaches instead of the proposed one? What will be the performance difference?*
>
> **Response:** We have not tried to compare the performance with other fusion approaches, as our main contribution is **not** proposing a structurally new fusion method, e.g., improving the bidirectional fusion module with an attention mechanism.  Instead, we explore point-to-pixel propagation with the goal of **directly enhancing the representation capability of the 3D LiDAR backbone** network by back-propagating gradients from the training objectives of the image branch. The minimalist point-to-pixel design best reveals this effect, which has never been identified previously to the best of our knowledge. The core ideas could be integrated with other fusion approaches in the future. But we believe our work provides valuable insights despite not comparing the performance with previous fusion methods.

---

> ### Comment · Area_Chair_vkAG · 2023-08-16
> **Rebuttal?**
>
> Dear reviewer,
>
> Can I ask you to please look at the rebuttal and see if it addresses your concerns?

---

> > ### Comment · Reviewer_s1rE · 2023-08-20
> >
> > The rebuttal resolves my concerns about the novelty of the proposed approach. However, the authors could not provide quantitative results to convince me about the performance part (e.g., weaker performance on the EASY category, fewer comparisons on other datasets, and with other fusion approaches ).

---

> > > ### Author Response · Authors · 2023-08-20
> > > **Thanks for your valuable feedback**
> > >
> > > Dear Reviewer s1rE,
> > >
> > > We deeply appreciate your valuable feedback and thoughtful examination of our work. We are delighted to learn that our rebuttal effectively addresses your concerns regarding the novelty issue. In response to your concerns about the quantitative results:
> > >
> > > **Performance on the EASY category:** While we acknowledge that our method may not achieve peak performance in every category, it is essential to recognize that the overall efficacy of a method is not solely dictated by its optimization in all metrics. We believe that the comprehensive performance of our approach, particularly in challenging scenarios, is indicative of its robustness and versatility.
> > >
> > > **Comparisons on other datasets:** With regard to the NuScenes dataset, we concur that there is room for additional benchmarking. However, the computational and resource constraints we faced during our experiments limited our explorations. It is noteworthy that our approach has shown promising results on benchmarks like KITTI and Waymo, which are well-regarded in the community. We believe that these results validate the effectiveness of our proposed method.
> > >
> > > **Comparisons with other fusion approaches:** As our main contribution is **not** proposing a **structurally** new fusion method, we have not tried to compare the performance with other fusion approaches. Our primary objective of the point-to-pixel module is to enhance the **representation capability** of the 3D LiDAR backbone network by leveraging gradients from the training objectives of the image branch. We experimentally demonstrated our claim in Table 2 of our manuscript. We posit the findings bring insights into the cross-modal knowledge distillation for 3D detection. Thus, the absence of a direct comparison with other fusion approaches **does not diminish** our contribution.
> > >
> > > We genuinely hope that these clarifications provide a clearer perspective on our research and its merits. Thanks again for your valuable time and feedback.
> > >
> > > Warm regards,
> > >
> > > Authors

---

### Official Review · Reviewer_5jwD · 2023-07-06

**Soundness:** 3 good
**Presentation:** 3 good
**Contribution:** 3 good
**Rating:** 5
**Confidence:** 4

**Summary:**

This paper propose a multi-modal usion-based 3d object detector named BiProDet. BiProDet adopts a bidirectional feature propagation mechanism, i.e.,  point-to-pixel module and pixel-to-point module. Besides, BiProDet propose a new auxiliary task called Normalized Local Coordinate (NLC) map.

**Strengths:**

The paper is well written, and the ablation study demonstrate the effectiveness of the proposed method.

**Weaknesses:**

1. Comparison with some closely related work is missing. For example, the proposed method is similar to EPNet++ ( [21] in the submitted manuscript) in several aspects. Both of them design bidirectional feature fusion and adopt semantic segmentation as an auxiliary task.
2. BEV representation is receiving increasing attention recently. However,  a comparison with these methods is missing.

**Questions:**

1. The proposed method is closely related to EPNet++. It would be better to provide a more detailed comparison between these two approaches.
2. BEV representation is receiving increasing attention recently and achieves sota results on many benchmarks. It would be better to compare the proposed method with BEV-base fusion methods (e.g. BEVFusion).
3. It would be more convincing to evaluate the proposed method on more datasets besides KITTI,  e.g., Nuscenes.
3. Why NLC is more beneficial for small objects such as ped. and cyc.?

**Limitations:**

The proposed fusion method is not as flexible as the BEV representation. And the comparison with BEV-based methods is missing.

---

> ### Author Rebuttal · Authors · 2023-08-09
>
> ### Response to Reviewer 5jwD
>
> We sincerely appreciate the time and effort you have dedicated to providing such insightful and comprehensive feedback on our work. The comments you have raised help us identify areas for improvement in our work. The review process has been informative in refining our paper.
>
> ### **Comment 1:** *Comparison with some closely related work is missing. It would be better to provide a more detailed comparison between the proposed method and EPNet++.*
>
> **Response:** Thanks for pointing out this related paper. It is also worth mentioning that the authors were **indeed** unaware of the preprint version of EPNet++ in Arxiv at the early stage of this project. Here we summarize the differences between our work and EPNet++:
> -  EPNet++ proposes an LI-Fusion layer to enable more interaction between the two modalities and finally obtains more comprehensive features. By contrast, we explore point-to-pixel propagation from the perspective of **directly improving the representation capability of the 3D LiDAR backbone** network. Specifically, by designing feasible 2D auxiliary tasks, the **gradients** back-propagated from the training objectives of the image branch can boost the representation ability of the point cloud backbone. We experimentally and exactly demonstrated our claim in Table 2 of our manuscript. To the best of our knowledge, it has never been identified previously that **2D** auxiliary tasks can be used to improve the **3D** backbone.
> -  We also verify that the bidirectional propagation benefits **not only** the 3D object detection task **but also** the 2D semantic segmentation task (see Table 7 of the supplementary material). The decent results show the potential of 2D-3D joint learning between 3D object detection and more 2D scene understanding tasks.
> -  We design a **concise yet effective** bidirectional propagation strategy without bells and whistles, which achieves significantly better performance than EPNet++ [4], according to its reported results on the KITTI test set. And we make the source code publicly available before EPNet++.
>
> ### **Comment 2:** *It would be better to compare the proposed method with BEV-base fusion methods. It would be more convincing to evaluate the proposed method on more datasets besides KITTI, e.g., Nuscenes.*
>
> **Response**: We appreciate the suggestion for including experiments on the NuScenes dataset. To be honest, it is very **difficult** to achieve competitive performance on the large-scale Nuscenes dataset in such a short time with limited GPU resources. Especially, the baseline **point-based** detector we used performs **poorly** on NuScenes or Waymo, which increases the experiment workload. Actually, when preparing this work, we have made great efforts to improve the performance of point-based detectors on the Waymo dataset to be **comparable** to the BEV-based detectors. Despite the missing experiments on NuScenes, we posit our method sheds light on point-based methods by offering new insights and an effective framework. Also, we believe the strong results on KITTI and Waymo sufficiently demonstrate the effectiveness of our proposed method and validate our motivation.
>
>
> ### **Comment 3:** *Why NLC is more beneficial for small objects such as ped. and cyc.?*
>
> **Response:** Insightful observation! Small objects like pedestrians tend to have **sparser** point clouds and be more sensitive to occlusion. While their LiDAR representations may be incomplete, the contour and appearance cues could still be clear in RGB images. By learning local spatial-aware features from images supervised by 2D NLC maps, we can better complement the sparse LiDAR observations for small objects. Thus, the local spatial information from images provides greater benefits for small objects with sparser point clouds.
>
> ### **Comment 4:** *The proposed fusion method is not as flexible as the BEV representation.*
>
> **Response:** While BEV methods show flexibility on large datasets, point-based approaches also have advantages, like **preserving** fine details without voxelization. In this work, we made great efforts to improve the performance of point-based detectors on the Waymo dataset to be comparable to the BEV-based detectors. For example, we found that it is crucial to develop a separate head for each category like CenterPoint, aiming to learn the biases of different categories and solve the category imbalance problem. We believe our method **sheds light on** point-based methods by offering new insights and an effective framework.

---

> ### Author Response · Authors · 2023-08-21
> **Gentle Reminder**
>
> Dear Reviewer 5jwD,
>
> Thank you for taking the time to review our submission and the favorable recommendation. As the discussion phase between the reviewers and authors is coming to an end, we would be grateful if you could acknowledge receipt of our responses and let us know if they address your concerns. We remain open and enthusiastic about any further discussions or clarifications you might deem necessary.
>
> Warm regards,
>
> Authors

---

### Official Review · Reviewer_gAXu · 2023-07-06

**Soundness:** 4 excellent
**Presentation:** 4 excellent
**Contribution:** 4 excellent
**Rating:** 7
**Confidence:** 5

**Summary:**

This work addresses the task of 3D object detection from LiDAR and cameras. Their main contribution is developing a joint 2D and 3D stream architecture, with simple bidirectional feature flow in the backbones. To improve this, they propose to predict NLC maps in the image stream. The proposed components demonstrate performance improvement on the KITTI dataset.

**Strengths:**

- The paper is clear and easy to read, with diagrams to explain crucial parts.
- The NLC prediction significantly improves performance, supporting the core hypothesis in the paper.
- As opposed to some existing works with joint 2D-3D backbones, the proposed bidirectional feature flow is simple without bells and whistles and complicated components.
- It is a non-trivial and interesting result that point-to-pixel flow during training only can still improve 3D detection performance, showing that 2D gradients can help 3D model learning.


**Weaknesses:**

- The statement “However, there is no evidence that these methods actually enhance the representation capability of the 3D LiDAR backbone network, which is also one of the most critical influencing factors.” in L41 - L46 (as well as its encompassing paragraph) is unclear to me and seems unsubstantiated. Many existing works (PointPainting, AutoAlign, etc) have enhanced the point cloud/3D voxels with 2D semantics before/while using the 3D backbone network and have demonstrated significant improvements in detection metrics by training end-to-end.
- I am slightly concerned that all of the ablative results shown are on the KITTI dataset, which is a small-scale dataset with low diversity. I would appreciate some ablations on at least the key components (NLC prediction, point-to-pixel flow) on the Waymo dataset. Nonetheless, it is impressive that a point-based method can achieve such strong performance on Waymo.
- Point-to-pixel flow during training only does improve 3D detection performance, which seems to demonstrate that training a 2D CNN can help the 3D network learn better features. However, this is slightly confounded with the possibility that *NLC* supervision is what is improving 3D detection when point-to-pixel flow is added, not the 2D CNN. It would strengthen the paper if point-to-pixel flow improves performance even when 3D NLC supervision (similar to that in PartA2) is done.


**Questions:**

- Please reference the weaknesses section for questions.
- I also want to ask if any augmentations are done in 2D. For instance, LR flipping may introduce ambiguity in predicting left vs right side of an object.


**Limitations:**

Authors do not appear to have included a limitations section.

---

> ### Author Rebuttal · Authors · 2023-08-09
>
> ### Response to Reviewer gAXu
> We sincerely appreciate the time and effort in evaluating our manuscript. Your meticulous review and thoughtful critiques truly reflect your deep domain expertise and diligence as a reviewer. In what follows, we will address your remaining concerns comprehensively and clearly.
>
> ### **Comment 1:** *The statement "there is no evidence that these methods actually enhance the representation capability of the 3D LiDAR backbone network" is not clear and seems unsubstantiated. Many existing works (PointPainting, AutoAlign, etc.) have enhanced the point cloud/3D voxels with 2D semantics before/while using the 3D backbone network and have demonstrated significant improvements.*
>
> **Response:** Sorry for the confusion caused. Prior works have indeed shown performance gains from **feature-level** fusion. By contrast, our key contribution is demonstrating that with feasible 2D auxiliary tasks, the **gradients** back-propagated from the training objectives of the image branch can boost the **representation ability** of the point cloud backbone. As analyzed in Table 2 of the manuscript, the mAP improves from 75.88% (Table2(a)) to 77.01% (Table2(b)) by adding the point-to-pixel module, precisely indicating the strengthened representation ability of the point cloud backbone because **only** the point cloud branch is used during inference. To the best of our knowledge, it has never been identified previously that **2D** auxiliary supervision tasks can be used to improve **3D** backbone.
>
> ### **Comment 2:** *I would appreciate some ablations on at least the key components (NLC prediction, point-to-pixel flow) on the Waymo dataset. Nonetheless, it is impressive that a point-based method can achieve such strong performance on Waymo.*
>
> **Response:** Thank you for your valuable comments. We agree with the reviewer that the ablation study will be more convincing on the Waymo dataset. As suggested, we conducted experiments on 20% Waymo training data, verifying consistent improvements from point-to-pixel propagation and 2D NLC supervision (see table below). We also appreciate you noting the strong Waymo results of our **point-based** approach. In fact, we made great efforts to improve the performance of point-based detectors on Waymo. And we found that it is **crucial** to develop a separate head for each category like CenterPoint, aiming to learn the biases of different categories and solve the category imbalance problem.
>
>
> Table T0: Effect of the key components in BiProDet on WOD val set. We report the results of APH on LEVEL 2.
>
> | Exp. | Point2Pixel | 2D NLC | Veh.   | Ped.   | Cyc.   | Mean      |
> | :--: | :-: | :----: | :----: | :----: | :----: | :-------: |
> | (a)  | &#10006;   | &#10006;      | 65\.67 | 57\.34 | 71\.64 | 64\.88 |
> | (b)  | &#10004;   | &#10006;      | 66\.67 | 58\.36 | 72\.39 | 65\.80 |
> | (c)  | &#10004;   | &#10004;       | 67\.75 | 59\.25 | 73\.9  | 66\.96 |
>
>
> ### **Comment 3:** *It is slightly confounded with the possibility that NLC supervision is what is improving 3D detection when point-to-pixel flow is added, not the 2D CNN. It would strengthen the paper if point-to-pixel flow improves performance even when 3D NLC supervision (similar to that in Part-A^2) is done.*
>
> **Response:** Thanks for your valuable suggestion. We have experimentally demonstrated the effectiveness of point-to-pixel propagation in Table 2 of our manuscript. Note the mAP is largely boosted from 75.88% (Table2(a)) to 77.01% (Table2(b)) with only a **single variable**, i.e., the point-to-pixel module is added. And following your suggestions, we further conducted the ablative experiment of point-to-pixel propagation when 2D NLC supervision is **replaced** with 3D NLC supervision. As shown in the table below, the point-to-pixel module improves the mAP consistently by 0.7%, validating its efficacy independent of NLCs supervision type.
>
> | Exp. | 3D NLC | P2I | Veh.   | Ped.   | Cyc.   | Mean   |
> | :--: | :----: | :-: | :----: | :----: | :----: | :----: |
> | (a)  | &#10004; | &#10006;   | 87\.09 | 63\.94 | 79\.91 | 76\.98 |
> | (b)  | &#10004;  | &#10004;   | 87\.23 | 65\.19 | 80\.62 | 77\.68 |
>
> ### **Comment 4:** *I also want to ask if any augmentations are done in 2D. For instance, LR flipping may introduce ambiguity in predicting left vs right side of an object.*
>
> **Response:** We appreciate you raising this important point. We do not perform any data augmentation on the image input. The LR flipping does introduce ambiguity in predicting the relative location of pixels inside an object.

---

> > ### Comment · Reviewer_gAXu · 2023-08-20
> >
> > I thank the authors for addressing all of my concerns. The proposed components demonstrate good improvement on the larger Waymo dataset as well, and the 2D gradient flow seems to improve 3D even without 3D NLC supervision, which is an interesting observation. As such, I maintain my original rating. I do hope that the authors could revise the manuscript to make Concern #1 more clear.

---

> > > ### Author Response · Authors · 2023-08-20
> > > **Response to Reviewer gAXu**
> > >
> > > Dear Reviewer gAXu,
> > >
> > > We are delighted to learn that our rebuttal effectively addresses your concerns. To address Concern #1, we will ensure that the necessary revisions are made to enhance clarity in the manuscript. Thanks again for your time and efforts in reviewing our manuscript.
> > >
> > > Warm regards,
> > >
> > > Authors

---

### Official Review · Reviewer_bgiL · 2023-07-09

**Soundness:** 3 good
**Presentation:** 3 good
**Contribution:** 2 fair
**Rating:** 5
**Confidence:** 5

**Summary:**

This paper proposes a method for multimodal (image and LiDAR) 3D object detection. The main purpose of the proposed method is to enhance the image branch. The authors design the task of NLC Map estimation, which is to predict the normalized local coordinates of points within a ground-truth box. The prediction happens on the image plane and the points are projected (carrying their NLC labels). In addition, image semantic segmentation is also adopted as an auxiliary task for the image branch. The image and LiDAR branch fuse their features during the forward of their backbones. The LiDAR branch predicts 3D instances based on the RPN proposals and RoI-pooled fused features.

**Strengths:**

1. Although the idea of predicting local coordinates within a bounding box is similar to Part-A^2, it is interesting to see that it can effectively enhance the image branch. I am also wondering if additionally introducing local coordinate prediction for the LiDAR branch can further boost performance.
2. The proposed method achieves SOTA results on the KITTI dataset, and the effectiveness of the auxiliary tasks are proved by ablation study.


**Weaknesses:**

1. The experiments are only conducted on the KITTI dataset, in fact, there are a bunch of recent multimodal 3D detection methods that provide results on the NuScenes dataset, so it would be more convincing with comparisons on NuScenes.
2. The bidirectional feature propagation module is simply gathering features from another modality by coordinate projection and feature transformation, which is not significantly novel.


**Questions:**

Would it be more straightforward to use local coordinate prediction in the LiDAR branch like Part-A^2? NLC estimation can also be performed for the points if they are not projected to the image plane?

**Limitations:**

The authors did not address the limitations.

---

> ### Author Rebuttal · Authors · 2023-08-09
>
> ### Response to Reviewer bgiL
>
> We sincerely appreciate the reviewer's time and effort in reviewing our paper. Thanks for your valuable comments and recognition of our work. In the following, we will comprehensively address your concerns.
>
> ### **Comment 1:** *The experiments are only conducted on the KITTI dataset. The results would be more convincing with comparisons on NuScenes.*
>
> **Response:** We appreciate your suggestion for conducting experiments on the NuScenes dataset. However, due to **limited** computational resources, achieving competitive performance on NuScenes in such a short time is very difficult. The baseline point-based detector (the LiDAR branch) we used performs **poorly** on NuScenes or Waymo, further increasing the experiment workload. Frankly speaking, when preparing this paper, we have made great efforts to improve the performance of point-based detectors on the Waymo dataset and achieve **comparable** accuracy to the BEV-based detectors. Despite the missing experiments on NuScenes, we posit our method **sheds light on** point-based methods by offering new insights and an effective framework. Also, we believe the strong results on KITTI and Waymo sufficiently demonstrate the effectiveness of our proposed method and validate our motivation.
>
> ### **Comment 2:** *The bidirectional feature propagation module is simply gathering features from another modality by coordinate projection and feature transformation, which is not significantly novel.*
>
> **Response:** The key motivation of our point-to-pixel module is **not** to propose structurally novel fusion mechanisms. Instead, we aim to **directly boost the representation capability of the point cloud backbone** network by back-propagating **gradients** from the training objectives of the image branch, due to the relatively weak representation capability of current point-based backbones for processing irregular 3D point clouds (see Lines 39-46 of the manuscript). We experimentally and exactly demonstrated our claim in Table 2 of our manuscript, where the mAP improves from 75.88% (Table2(a)) to 77.01% (Table2(b)) by adding the point-to-pixel module, indicating strengthened representation ability of the point cloud backbone. To our knowledge, exploiting **2D** auxiliary tasks to improve the representation capability of **3D** backbones has not been identified before. The concise point-to-pixel design best demonstrates this capability. Besides, **Reviewer gAXu** also acknowledged such a manner is non-trivial and interesting.
>
> ### **Comment 3:** *Would it be more straightforward to use local coordinate prediction in the LiDAR branch like Part-A^2?*
>
> **Response:** Thank you for the valuable suggestion. We conducted experiments by predicting NLCs directly in the LiDAR branch, which improves mAP from 77.18% to 77.68% (see table below). By contrast, supervising 2D NLC map estimation further boosts mAP to 78.64%, showing superiority in exploiting local spatial-aware features from the image branch to complement sparse point cloud representations. The results validate the advantage of our design.
>
>
> | Setting | Car    | Ped.   | Cyc.   | mAP    |
> | :------ | :----- | :----- | :----- | :----- |
> | w/o NLC | 87\.10 | 64\.52 | 79\.94 | 77\.18 |
> | 3D NLC  | 87\.23 | 65\.19 | 80\.62 | 77\.68 |
> | 2D NLC  | 87\.18 | 67\.52 | 81\.21 | 78\.64 |

---

> > ### Comment · Reviewer_bgiL · 2023-08-21
> >
> > The authors' rebuttal roughly addressed my concerns, and I maintain my previous rating.

---

> ### Author Response · Authors · 2023-08-21
> **Gentle Reminder**
>
> Dear Reviewer bgiL,
>
> Thank you for taking the time to review our submission and the favorable recommendation. As the discussion phase between the reviewers and authors is coming to an end, we would be grateful if you could acknowledge receipt of our responses and let us know if they address your concerns. We remain open and enthusiastic about any further discussions or clarifications you might deem necessary.
>
> Warm regards,
>
> Authors

---

### Author Rebuttal · Authors · 2023-08-09

### General Response
We thank all reviewers for your time and constructive comments. Here we want to summarize a few key clarifications concerning the contributions of our work again:

**(1) The novelty of our work.** The key motivation of our point-to-pixel module is not to propose structurally novel fusion mechanisms. Instead, we aim to **directly boost the representation capability of the point cloud backbone** network by back-propagating **gradients** from the training objectives of the image branch, due to the relatively weak representation capability of current point-based backbones for processing irregular 3D point clouds (see Lines 39-46 of the manuscript). We experimentally demonstrated our claim in Table 2 of our manuscript, where the mAP improves from 75.88% (Table2(a)) to 77.01% (Table2(b)) by adding the point-to-pixel module, **precisely** indicating the strengthened representation ability of the point cloud backbone because **only** the point cloud branch is used during inference. To our knowledge, exploiting **2D** auxiliary tasks to improve **3D** backbones has not been identified before. The concise point-to-pixel design best demonstrates this capability.

It would be better to **holistically** understand the two key components, i.e., the NLC map estimation and point-to-pixel module, rather than in isolation. On one hand, we can learn local spatial-aware information from the image branch with NLC map supervision. On the other hand, we demonstrate that the gradients back-propagated from the image branch through the point-to-pixel module can boost the representation ability of the point cloud backbone. The suitable 2D tasks and the point-to-pixel module are **intricately** linked, and it is only by combining them that the highest detection accuracy can be achieved.

Besides, we demonstrate that the 2D-3D joint learning paradigm benefits **not only** the 3D object detection task **but also** the 2D semantic segmentation task (see Table 7 of the supplementary material). The point features can naturally complement RGB image features by providing 3D geometry and semantics, which are robust to illumination changes and help distinguish different classes of objects for 2D visual information. The results suggest the potential of joint training between 3D object detection and more 2D scene understanding tasks in autonomous driving.

**(2) Experiments on the NuScenes dataset.** Due to limited computational resources, achieving competitive performance on NuScenes in such a short time is very difficult. The employed baseline **point-based** detector (the LiDAR branch) performs **poorly** on NuScenes or Waymo, increasing the experiment workload. Frankly speaking, when preparing this paper, we have made great efforts to improve the performance of point-based detectors on the Waymo dataset to **comparable** accuracy to the BEV-based detectors. Despite the missing experiments on NuScenes, we posit our method **sheds light on** point-based methods by offering new insights and an effective framework. In conclusion, we believe the strong results on KITTI and Waymo sufficiently demonstrate the effectiveness of our proposed method and validate our motivation.

 We will make the reviews and author discussion public. Besides, we will include the newly added experiments and analysis in the final manuscript/supplementary material.

---

> ### Author Response · Authors · 2023-08-16
> **Looking forward to hearing from you**
>
> Dear Reviewers,
>
> Thank you for taking the time to review our submission and providing us with constructive comments. We would like to inquire if our responses have adequately addressed your earlier concerns. Additionally, if you have any further concerns or suggestions, we would be more than happy to address and discuss them to enhance the quality of the paper. We eagerly await your response and look forward to hearing from you.
>
> Best regards,
>
> The authors

---

### Decision · Program_Chairs · 2023-09-21

**Decision:**

Accept (poster)

**Comment:**

All but one reviewer vote for acceptance. Reviewers praise the simplicity and effectiveness of the proposed approach. This AC votes for acceptance, but urges the authors to add comparisons on additional datasets as requested by multiple reviewers.